# Generalizable Spectral Embedding with an Application to UMAP

**Nir Ben-Ari**                                                                *nirnirba@gmail.com*
*Department of Computer Science*
*Bar-Ilan University*

**Amitai Yacobi**                                                          *amitaiyacobi@gmail.com*
*Department of Computer Science*
*Bar-Ilan University*

**Uri Shaham**                                                              *uri.shaham@biu.ac.il*
*Department of Computer Science*
*Bar-Ilan University*

**Reviewed on OpenReview:** *https://openreview.net/forum?id=8cuQwztCKk*

## Abstract

Spectral Embedding (SE) is a popular method for dimensionality reduction, applicable across diverse domains. Nevertheless, its current implementations face three prominent drawbacks which curtail its broader applicability: generalizability (i.e., out-of-sample extension), scalability, and eigenvectors separation. Existing SE implementations often address two of these drawbacks; however, they fall short in addressing the remaining one. In this paper, we introduce *Sep-SpectralNet* (eigenvector-separated SpectralNet), a SE implementation designed to address *all* three limitations. Sep-SpectralNet extends SpectralNet with an efficient post-processing step to achieve eigenvectors separation, while ensuring both generalizability and scalability. This method expands the applicability of SE to a wider range of tasks and can enhance its performance in existing applications. We empirically demonstrate Sep-SpectralNet's ability to consistently approximate and generalize SE, while maintaining SpectralNet's scalability. Additionally, we show how Sep-SpectralNet can be leveraged to enable generalizable UMAP visualization.[1]

## 1 Introduction

Spectral Embedding (SE) is a popular non-linear dimensionality reduction method (Belkin & Niyogi, 2003; Coifman & Lafon, 2006b), finding extensive utilization across diverse domains in recent literature. Notable applications include UMAP (McInnes et al., 2018) (the current state-of-the-art visualization method), Graph Neural Networks (Zhang et al., 2021; Beaini et al., 2021; Defferrard et al., 2016), signal propagation on graphs (Park et al., 2022; Klaine et al., 2017) and analysis of proteins (Campbell et al., 2015; Zhu & Schlick, 2021). The core of SE involves a projection of the samples into the space spanned by the leading eigenvectors of the Laplacian matrix (i.e., those corresponding to the smallest eigenvalues), derived from the pairwise similarities between the samples. It is an expressive method which is able to preserve the global structure of high-dimensional input data, underpinned by robust mathematical foundations (Belkin & Niyogi, 2003; Katz et al., 2019; Lederman & Talmon, 2018; Ortega et al., 2018).

Despite the popularity and significance of SE, current implementations suffer from three main drawbacks: (1) *Generalizability* - the ability to directly embed a new set of test points after completing the computation

---

[1]Sep-SpectralNet: `https://github.com/shaham-lab/GrEASE`; NUMAP: `https://github.com/shaham-lab/NUMAP`

on a training set (i.e., out-of-sample extension); (2) *Scalability* - the ability to handle a large number of samples within a reasonable time-frame; (3) *Eigenvectors separation* - the ability to output the *basis* of the leading eigenvectors $(v_2, \ldots, v_{k+1})$, rather than only the space spanned by them. These three properties are crucial for modern applications of SE in machine learning. While most SE implementations address two of these three limitations, they often fall short in addressing the remaining one (see Tab. 1 and Sec. 2).

Notably, eigenvector separation has attracted considerable attention in recent years (Pfau et al., 2018; Gemp et al., 2020; Deng et al., 2022; Lim et al., 2022). This work focuses on this property, which is a key aspect underlying applications such as Fiedler vector, Diffusion maps, and Diffusion analysis. The latter two involve examining the evolution of random walks on graphs across different time scales. This framework enables a wide range of applications: analyzing molecular transitions (Glielmo et al., 2021; Chiavazzo et al., 2017), sampling rare protein transitions (Ghamari et al., 2024), uncovering latent structures in disordered materials (Hardin et al., 2024), studying the global organization of cortical features in various neuropsychiatric conditions (Park et al., 2021; 2022; Dong et al., 2020), supporting enhanced-sampling techniques (Zheng et al., 2011; 2013), and facilitating dimensionality reduction in self-organizing networks (Klaine et al., 2017). Eigenvector separation is required for those applications to enable scaling each diffusion component.

Shaham et al. (2018) presented SpectralNet, which tackles the scalability and generalizability limitations of Spectral Clustering (SC), a key application of SE. However, we prove that due to a rotation and reflection ambiguity in its loss function, SpectralNet cannot directly be adapted for SE in general, as it cannot separate the eigenvectors. In this paper, we present an eigendecomposition based post-processing procedure to resolve the eigenvectors separation issue in SpectralNet, thereby, extending SpectralNet into a scalable and generalizable implementation of SE, which we call *Sep-SpectralNet* (eigenvector-separated SpectralNet). Sep-SpectralNet's ability to separate the eigenvectors, while maintaining the generalizability and scalability of SpectralNet, offers a pathway to enhance numerous existing applications of SE, that require eigenvector separation, and provides a foundation for developing new applications.

One such application is UMAP (McInnes et al., 2018), a widely used visualization method whose ability to preserve global structure relies heavily on its Spectral Embedding (SE) initialization (Kobak & Linderman, 2021). Although Parametric UMAP (P. UMAP) was proposed to address UMAP's lack of generalizability (Sainburg et al., 2021), it omits SE, limiting its ability to replicate UMAP's structural fidelity. Despite this limitation, P. UMAP has gained traction in various domains (Xu & Zhang, 2023; Eckelt et al., 2023; Leon-Medina et al., 2021; Xie et al., 2024; Yoo et al., 2022), underscoring the need for a generalizable alternative that retains UMAP's structural strengths. To this end, we introduce an extension of Sep-SpectralNet that integrates SE-based initialization with the UMAP loss, named NUMAP, preserving global structure while enabling generalization. We show that eigenvector separation is required for NUMAP's performance. As shown empirically (Sec. 5), this extension enhances UMAP's applicability to dynamic settings such as online learning and time-series data visualization.

Our key contributions are: (1) We introduce Sep-SpectralNet, a novel approach for approximate Spectral Embedding (SE) that jointly addresses scalability, generalizability, and eigenvector separation. (2) We present an application of Sep-SpectralNetto generalizable UMAP. (3) We propose a new evaluation method to measure global structure preservation.

## 2 Related Work

Current SE implementations typically address two out of its three primary limitations: generalizability, scalability, and eigenvector separation (Tab. 1). Below, we outline key implementations that tackle each pair of these challenges. Following this, we discuss recent works related to eigenvectors separation and generalizable visualizations techniques.

**Scalable with eigenvectors separation.** Popular implementations of SE are mostly based on sparse matrix decomposition techniques (e.g., ARPACK (Lehoucq et al., 1998), AMG (Brandt et al., 1984), LOBPCG (Benner & Mach, 2011)). These methods are relatively scalable, as they are almost linear in the number of samples. Nevertheless, their out-of-sample extension is far from trivial. Usually, it is done by out-of-sample extension (OOSE) methods such as Nyström (Nyström, 1930) or Geometric Harmonics (Coifman & Lafon,

| Method | Generalizability | Scalability | Eigenvector Separation |
|---|:---:|:---:|:---:|
| LOBPCG | ✗ | ✓ | ✓ |
| SpectralNet | ✓ | ✓ | ✗ |
| DiffusionNet | ✓ | ✗ | ✓ |
| **Sep-SpectralNet (ours)** | ✓ | ✓ | ✓ |

Table 1: **Sep-SpectralNet is the only method to have the three desired properties of SE implementation.** Comparison between key SE methods via their ability to generalize to unseen samples, scale to large datasets and separate the eigenvectors.

2006a; Lafon et al., 2006). However, these methods provide only local extension (i.e., near existing training points), and are both computationally and memory restrictive, as they rely on computing the distances between every new test point and all training points.

**Scalable and generalizable.** Several approaches to spectral clustering (SC) approximate the space spanned by the first eigenvectors of the Laplacian matrix, which is sufficient for clustering purposes, and can also benefit other specific applications. For example, SpectralNet (Shaham et al., 2018) leverages deep neural networks to approximate the first eigenfunctions of the Laplace-Beltrami operator in a scalable manner, thus also enabling fast inference of new unseen samples. BASiS (Streicher et al., 2023) achieves these goals using affine registration techniques to align batches. However, these methods' inability to separate the eigenvectors prevents their use in many modern applications.

**Generalizable with eigenvectors separation.** Another proposed approach to SE is DiffusionNet (Mishne et al., 2019), a deep-learning framework for generalizable Diffusion Maps embedding (Coifman & Lafon, 2006b), which is similar to SE. However, the training procedure of the network is computationally expensive, therefore restricting its usage for large datasets.

In contrast, we introduce Sep-SpectralNet, which addresses all three limitations - generalizes the separated eigenvectors to unseen points with a single feed-forward operation, while maintaining SpectralNet's scalability.

**Eigenvectors separation.** Extensive research has been conducted on the eigenvectors separation problem, both within and beyond the spectral domain (Lim et al., 2022; Ma et al., 2024). Some rotation criteria such as ICA and VARIMAX are well known, but regarding the spectral domain, they do not yield the natural separation, i.e., the true eigenvectors. Recent spectral approaches remain constrained computationally, both by extensive run-time and memory consumption. For example, Pfau et al. (2018) proposed a solution to this issue by masking the gradient information from the loss function. However, this approach necessitates the computation of full Jacobians at each time step, which is highly computationally intensive. Gemp et al. (2020) employs an iterative method to learn each eigenvector sequentially. Namely, they learn an eigenvector while keeping the others frozen. This process has to be repeated $k$ times (where $k$ is the embedding dimension), which makes this approach also computationally expensive. Deng et al. (2022) proposed an improvement of the latter, by parallel training of $k$ NNs. However, as discussed in their paper, this approach becomes costly for large values of $k$. Furthermore, it necessitates retaining $k$ trained networks in memory, which leads to significant memory consumption. Chen et al. (2022) proposed a post-processing solution to this problem using the Rayleigh-Ritz method. However, this approach involves the storage and multiplication of very large dense matrices, rendering it impractical for large datasets. In contrast, Sep-SpectralNet offers an efficient one-shot post-processing solution to the eigenvectors separation problem.

**Generalizable visualization.** Several works have attempted to develop parametric approximations for non-parametric visualization methods, in addition to Parametric UMAP (P. UMAP) (Sainburg et al., 2021). Notable examples include (Van Der Maaten, 2009), (Kawase et al., 2022) and (Damrich et al., 2022), which

use NNs to make t-SNE generalizable, and (Schofield & Lensen, 2021), which aims to make UMAP more interpretable. However, P. UMAP has demonstrated superior performance. NUMAP presents a method to surpass P. UMAP in terms of global structure preservation.

## 3 Preliminaries

In this section, we begin by providing the fundamental definitions that will be used throughout this work. Additionally, we briefly outline the key components of UMAP and P. UMAP.

### 3.1 Spectral Embedding

Let $\mathcal{X} = \{x_1, \ldots, x_n\} \subseteq \mathbb{R}^d$ denote a collection of unlabeled data points drawn from some unknown distribution $\mathcal{D}$. Let $W \in \mathbb{R}^{n \times n}$ be a positive symmetric graph affinity matrix, with nodes corresponding to $\mathcal{X}$, and let $D$ be the corresponding diagonal degree matrix (i.e. $D_{ii} = \sum_{j=1}^{n} W_{ij}$). The Unnormalized Graph Laplacian is defined as $L = D - W$. Other normalized Laplacian versions are the Symmetric Laplacian $L_{\text{sym}} = D^{-\frac{1}{2}} L D^{-\frac{1}{2}}$ and the Random-Walk (RW) Laplacian $L_{\text{rw}} = D^{-1} L$. Sep-SpectralNet is applicable to all of these Laplacian versions. The eigenvalues of $L$ can be sorted to satisfy $0 = \lambda_1 \leq \lambda_2 \leq \cdots \leq \lambda_n$ with corresponding eigenvectors $v_1, \ldots, v_n$ (Von Luxburg, 2007). It is important to note that the first pair (i.e., $\lambda_1, v_1$) is trivial - for every Laplacian matrix $\lambda_1 = 0$, and for the unnormalized and RW Laplacians $v_1 = \frac{1}{\sqrt{n}} \vec{1}$, namely the constant vector.

For a given target dimension $k$, the first non-trivial $k$ eigenvectors provide a natural non-linear low-dimensional embedding of the graph which is known as *Spectral Embedding* (SE). In practice, we denote by $V \in \mathbb{R}^{n \times k}$ the matrix containing the first non-trivial $k$ eigenvectors of the Laplacian matrix as its columns (i.e., $v_2, \ldots, v_{k+1}$). The SE representation of each sample $x_i \in \mathbb{R}^d$ is the $i$th row of $V$, i.e., $y_i = (v_2(i), \ldots, v_{k+1}(i))$.

### 3.2 SpectralNet

A prominent method for addressing scalability and generalizability in Spectral Clustering (SC) is using deep neural networks, for example SpectralNet (Shaham et al., 2018). SpectralNet follows a common methodology for transferring the problem of matrix decomposition to its smallest eigenvectors to an optimization problem, through minimization of the Rayleigh Quotient (RQ).

**Definition 1.** *The Rayleigh quotient (RQ) of a Laplacian matrix $L \in \mathbb{R}^{n \times n}$ is a function $R_L : \mathbb{R}^{n \times k} \to \mathbb{R}$ defined on $A \in \mathbb{R}^{n \times k}$ by*

$$R_L(A) = \text{Tr}(A^T L A),$$

SpectralNet first minimizes the RQ on small batches, while enforcing orthogonality. Namely, it approximates $\theta^*$ which minimizes

$$\mathcal{L}_{\text{spectralnet}}(\theta) = \frac{1}{m^2} R_L\big(f_\theta(X)\big) \quad s.t. \quad \frac{1}{m} f_\theta(X)^T f_\theta(X) = I_{k \times k}, \tag{1}$$

where $m$ is the minibatch size and $X \in \mathbb{R}^{m \times d}$ contains the $d$-dimensional data samples as rows. Thereby, it learns a map $f : \mathbb{R}^d \to \mathbb{R}^k$ (where $d$ is the input dimension) which approximates the space spanned by the first $k$ eigenfunctions of the Laplace-Beltrami operator on the underlying manifold $\mathcal{D}$ (Belkin & Niyogi, 2006; Shi, 2015). Then, it clusters the representations via $k$-means. These eigenfunctions are a natural generalization of the SE to unseen points, enabling both scalable and generalizable spectral clustering.

### 3.3 UMAP and Parametric UMAP

UMAP (McInnes et al., 2018) is a widely used visualization method, known for its scalability and its ability to preserve global structure. It constructs a graph from high-dimensional data and learns a low-dimensional representation by minimizing the KL-divergence between the input graph and the representation graph.

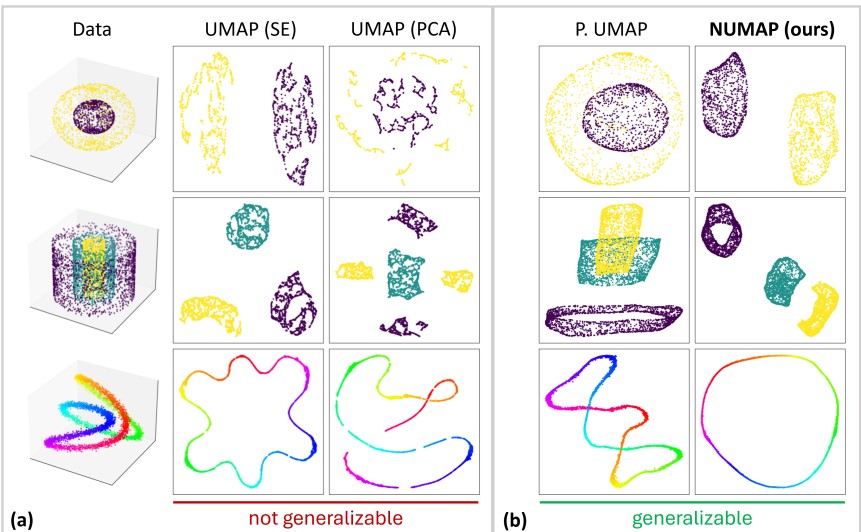

Figure 1: **Spectral Embedding is crucial for global structure preservation.** (a) Non-parametric UMAP preserves the global structure when initialized with SE, but fails to do so with PCA initialization - note the in-cluster separation in the middle row; (b) A similar effect is observed in generalizable implementations: P. UMAP, which does not involve SE, , fails to maintain global structure, resulting in overlapping clusters (e.g., first and second rows). In contrast, NUMAP (ours) achieves global structure preservation comparable to UMAP while supporting generalizability.

The UMAP method can be divided into three components (summarized in Fig. 3): (1) constructing a graph which best captures the global structure of the input data; (2) initializing the representations via SE; (3) Learning the representations, via SGD, which best capture the original graph. This setup does not facilitate generalization, as both steps (2) and (3) lack generalizability.

Additionally, as discussed in (Kobak & Linderman, 2021), UMAP primarily derives its global preservation abilities, as well as its consistency, from initializing the representations using SE. Therefore, the SE initialization serves as a critical step for UMAP to uphold the global structure (see demonstrations in Fig. 1a). Global preservation, in this context, refers to the separation of different classes, and avoiding the separation of existing classes. We refer the reader to (Kobak & Linderman, 2021) for a more comprehensive discussion about the effects of informative initialization on UMAP's performance.

Recently, a generalizable version of UMAP, known as Parametric UMAP (P. UMAP), was introduced (Sainburg et al., 2021). P. UMAP replaces step (3) with the training of a neural network. Importantly, it overlooks step (2), the SE initialization. Consequently, P. UMAP may struggle to preserve global structure, particularly when dealing with non-linear structures. Fig. 1b illustrates this phenomenon using several simple yet non-linear structures. These examples are particularly insightful for highlighting the importance of SE in preserving global structure, as the expected outcome of a good visualization is known. Noticeably, P. UMAP fails to preserve global structure (e.g., it does not separate different clusters).

## 4 Method

### 4.1 Motivation

It is well known that $V \in \mathbb{R}^{n \times k}$, whose columns are the first $k$ eigenvectors of $L$ (i.e., those corresponding to the $k$ smallest eigenvalues), minimizes $R_L(A)$ under orthogonality constraint (i.e., $A^T A = I$) (Li, 2015).

However, a rotation and reflection ambiguity of the RQ prohibits a trivial adaptation of this concept to SE. Basic properties of trace imply that for any orthogonal matrix $Q \in \mathbb{R}^{k \times k}$ the matrix $U := VQ$ satisfies

$R_L(U) = R_L(V)$. Thus, every such $U$ also minimizes $R_L$ under the orthogonality constraint, and therefore this kind of minimization solely is missing eigenvectors separation, which is crucial for many applications.

In fact, as stated in Lemma 1, the aforementioned form $VQ$ is the only form of a minimizer of $R_L$ under the orthogonality constraint. For conciseness, we provide our proof to the lemma in App. A.

**Lemma 1.** *Every minimizer of $R_L$ under the orthogonality constraint, is of the form $VQ$, where $V$ is the first $k$ eigenvectors matrix of $L$ and $Q$ is an arbitrary squared orthogonal matrix.*

An immediate result of Lemma 1 is that SpectralNet's method, using a deep neural network for RQ minimization (while enforcing orthogonality), does not lead to the SE. However, it only leads to the space spanned by the constant vector and the leading $k-1$ eigenvectors of $L$, with different rotations and reflections for each run. Therefore, each time the RQ is minimized, it results in a different linear combination of the smallest eigenvectors. Although this is sufficient for clustering purposes, as we search for reproducibility, consistency, and separation of the eigenvectors, the RQ cannot solely provide the SE, necessitating the development of new techniques in Sep-SpectralNet.

### 4.2 Sep-SpectralNet

**Setup.** Here we present the two key components of Sep-SpectralNet, a scalable and generalizable SE method. We consider the following setup: Given a training set $\mathcal{X} \subseteq \mathbb{R}^d$ and a target dimension $k$, we construct an affinity matrix $W$, and compute an approximation of the leading eigenvectors of its corresponding Laplacian. In practice, we first utilize SpectralNet (Shaham et al., 2018) to approximate the space spanned by the first $k+1$ eigenfunctions of the corresponding Laplace-Beltrami operator, and then find each of the $k$ leading eigenfunctions within this space (i.e. the SE). Namely, Sep-SpectralNet computes a map $F_\theta : \mathbb{R}^d \to \mathbb{R}^k$, which approximates the map $\bar{f} = (f_2, \dots, f_{k+1})$, where $f_i$ is the $i$th eigenfunction of the Laplace-Beltrami operator on the underlying manifold $\mathcal{D}$.

**Eigenspace approximation.** As empirically showed in (Shaham et al., 2018), and motivated from Lemma 1, SpectralNet loss is minimized when $F_\theta = T \circ (f_1, \dots, f_{k+1})$, where $T : \mathbb{R}^{k+1} \to \mathbb{R}^{k+1}$ is an arbitrary isometry. That is, $F_\theta$ approximates the space spanned by the first $k+1$ eigenfunctions. However, the SE (i.e. each of the leading eigenfunctions) is poorly approximated. Each time the RQ is minimized, the eigenfunctions are approximated up to a different isometry $T$. Fig. 2a demonstrates this phenomenon on the toy moon dataset - a noisy half circle linearly embedded into 10-dimension input space (see Sec. 5.1). Employing SpectralNet approach indeed enables us to consistently achieve a perfect approximation of the space (i.e., the errors at the left histograms are accumulated around 0). However, when comparing vector to vector, it becomes apparent that the SE was seldom attained.

**SE approximation.** To separate the eigenvectors, and thereby consistently get the SE, we present a simple use of Lemma 1. Notice that based on Lemma 1 we can compute a rotated version of the diagonal eigenvalues matrix. Namely,

$$(VQ)^T L(VQ) = Q^T V^T L V Q = Q^T \Lambda Q =: \tilde{\Lambda},$$

where $\Lambda$ is the diagonal eigenvalues matrix. Due to the uniqueness of eigendecomposition, the eigenvectors and eigenvalues of the small matrix $\tilde{\Lambda} \in \mathbb{R}^{k+1 \times k+1}$ are $Q^T$ and $\mathrm{diag}(\Lambda)$, respectively. Hence, by diagonalizing $\tilde{\Lambda}$ we get the eigenvalues and are also able to separate the eigenvectors (i.e., approximate the SE).

In practice, as $Q$ is a property of SpectralNet optimization (manifested by the parameters), we compute the matrix $\tilde{\Lambda}$ by averaging over a few random minibatches and diagonalize it. Thereby, making this addition very cheap computationally. The eigenvectors matrix of $\tilde{\Lambda}$ is the inverse of the orthogonal matrix $Q$, and hence by multiplying the output of the learned map $F_\theta$ by this matrix, the SE is retained. Also, the eigenvalues of $\tilde{\Lambda}$ are the eigenvalues of $L$.

This step is simple and elegant, yet its impact is significant. The effect of this intentional rotation is represented in the Fig. 2a. Sep-SpectralNet is not only able to consistently approximate the space, but also to approximate each eigenvector.

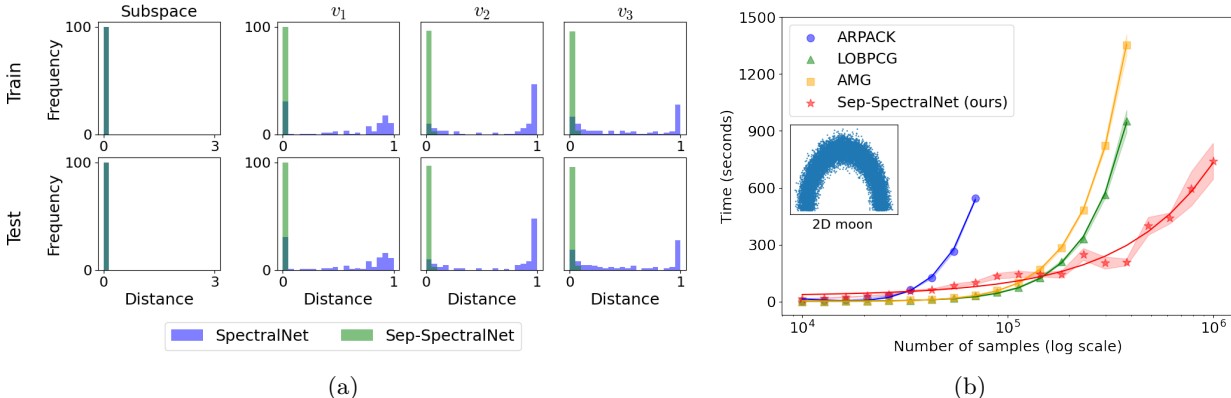

(a)                                                                    (b)

Figure 2: **(a) Sep-SpectralNet separates the eigenvectors.** Approximation of the 2-dimensional SE of the moon dataset (a 2D moon linearly embedded into 10D input space) using SpectralNet (in blue) and Sep-SpectralNet (in green) over 100 runs, on train (top row) and test (bottom row). Left column: distribution of the Grassmann distance between the output and true subspace. Second to Fourth columns: distribution of the $\sin^2$ distance between each output and true eigenvector separately. Evidently, while SpectralNet's errors are distributed over a large range of values, Sep-SpectralNet's errors are small, capturing only the smallest error bin in the figure, indicating on eigenvector separation. **(b) Sep-SpectralNet's training is as scalable non-generalizable spectral methods.** Running times of SE using Sep-SpectralNet vs. other methods on the moon dataset, relative to the number of samples, and with standard deviation confidence intervals. Evidently, Sep-SpectralNet is the fastest asymptotically.

**Algorithms Layout.** Our end-to-end training approach is summarized in Algorithms 1 and 2 in App. B. We first train $F_\theta$ to approximate the first eigenfunctions up to isometry (Algorithm 1) (Shaham et al., 2018), then compute $Q^T$ and $\Lambda$ to separate the eigenvectors and recover the SE and corresponding eigenvalues (Algorithm 2). App. C details additional considerations about the implementation.

Once we have $F_\theta$ and $Q^T$, computing the embeddings of the train set or of new test points (i.e., out-of-sample extension) is straightforward: we simply propagate each test point $x_i$ through the network $F_\theta$ to obtain their embeddings $\tilde{y}_i$, and use $Q^T$ to get the SE embeddings $y_i = \tilde{y}_i Q^T$.

**Time and Space complexity.** As the network iterates over small batches, and the post-processing operation is much cheaper, Sep-SpectralNet's time complexity is approximately linear in the number of samples. This is also demonstrated in Fig. 2b, where the continuous red line, representing linear regression, aligns with our empirical results. App. C provides a discussion about the complexity of Sep-SpectralNet. Sep-SpectralNet is also highly memory-efficient, requiring only minibatch-sized graphs or matrices in memory, rather than the full graph.

## 4.3 NUMAP

Here, we demonstrate Sep-SpectralNet's extension to UMAP, one of many methods that may benefit from a generalizable SE. As discussed in Sec. 3.3, the SE initialization is crucial for the global preservation abilities of UMAP. Therefore, we seek a method to incorporate SE into a generalizable version of UMAP. A naïve approach would be to fine-tune Sep-SpectralNet using UMAP loss (Eq. 2). However, during this implementation, we encountered the phenomenon of catastrophic forgetting (see App. E).

The core of our idea is illustrated in Fig. 3. Initially, we use Sep-SpectralNet to learn a parametric representation of the $k$-dimensional SE of the input data. Subsequently, we train an NN to map from the SE to the final $\ell$-dimensional embedding space (usually for $\ell = 2, 3$), utilizing UMAP contrastive loss. The objective of the second NN is to identify representations that best capture the local structure of the input data graph. SE transforms complex non-linear structures into simpler linear structures, allowing the second NN to preserve

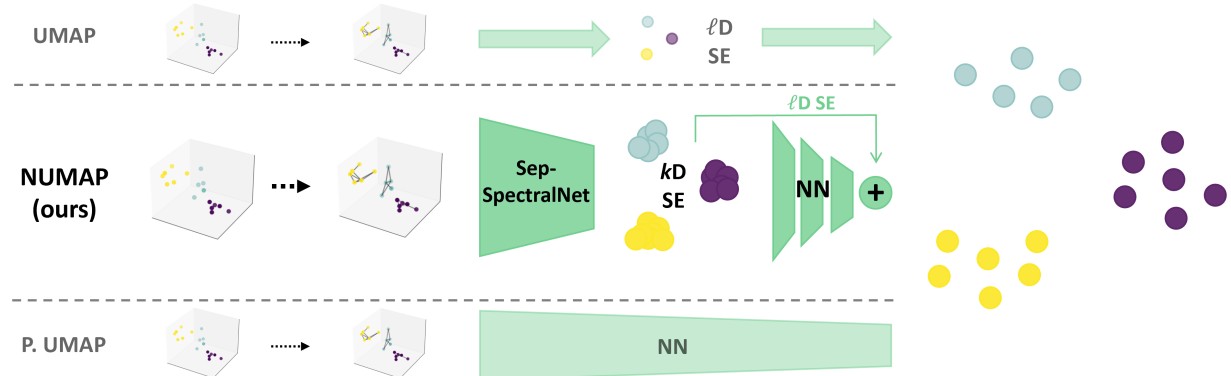

Figure 3: **Incorporating Sep-SpectralNet for generalizable UMAP.** UMAP vs. NUMAP vs. P. UMAP overview. NUMAP integrates SE, as in UMAP, while enabling generalization. First, it learns a parametric representation of the $k$-dimensional SE. Then, it learns a map from the SE to the final $\ell$-dimensional space, incorporating residual connections from the first $\ell$ input components to the output layer.

both local and global structures effectively. To enhance this capability, we incorporate residual connections in the second NN, $G_\theta$, which link the first $\ell$ components of the input (the $\mathrm{SE}_{1:\ell}$) directly to the output. Thus, the NUMAP embedding is given by $Y = \mathrm{SE}(X)_{1:\ell} + G_\theta(X)$, that minimizes the UMAP's loss

$$\sum_{e \in E} w_h(e) \log\left(\frac{w_h(e)}{w_l(e)}\right) + (1 - w_h(e)) \log\left(\frac{1 - w_h(e)}{1 - w_l(e)}\right), \tag{2}$$

where $w_h(e), w_l(e)$ are the corresponding weights of the edge $e$ in the high-dimensional input ($X$) graph and the low-dimensional output ($Y$) graph, respectively. It should be noted that this could not have been possible without Sep-SpectralNet's ability to separate the eigenvectors, as the first $\ell$ components of SpectralNet's output are merely arbitrary linear combinations of the $k$-dimensional SE. Also, it would not be practical without Sep-SpectralNet's inherent generalizability and scalability. Fig. 1 demonstrates this capability with several simple structures.

### 4.4 Additional Applications

In this section we seek to highlight Sep-SpectralNet's potential impact on important tasks and applications (besides generalizable UMAP), as it integrates generalizability, scalability and eigenvectors separation. As discussed in Sec. 1, SE is applied across various domains, many of which can benefit generalizability capabilities by replacing the current SE implementation with Sep-SpectralNet. We therefore elaborate herein the significance of SE in selected applications, and discuss how Sep-SpectralNet, as a generalizable approximation of it, can enhance their effectiveness and applicability.

**Fiedler vector and value.** A special case of SE is the Fiedler vector and value (Fiedler, 1973; 1975). The Fiedler value, also known as algebraic connectivity, refers to the second eigenvalue of the Laplacian matrix, while the Fiedler vector refers to the associated eigenvector. This value quantifies the connectivity of a graph, increasing as the graph becomes more connected. Specifically, if a graph is not connected, its Fiedler value is 0. The Fiedler vector and value are a main topic of many works (Andersen et al., 2006; Barnard et al., 1993; Kundu et al., 2004; Shepherd et al., 2007; Cai et al., 2018; Zhu & Schlick, 2021; Tam & Dunson, 2020). As Sep-SpectralNet is able to distinguish between the eigenvectors and approximate the eigenvalues, it has the capability to approximate both the Fiedler vector and value, while also generalizing the vector to unseen samples (see Sec. 5.1).

**Diffusion Maps.** A popular method which incorporates SE, alongside the eigenvalues of the Laplacian matrix, is Diffusion Maps (Coifman & Lafon, 2006b). Diffusion Maps embeds a graph (or a manifold) into a

space where the pairwise Euclidean distances are equivalent to the pairwise Diffusion distances on the graph. This approach is widely used (e.g., (Ghamari et al., 2024; Hardin et al., 2024; Park et al., 2021)).

In practice, for an $k$-dimensional embedding space and a given $t \in \mathbb{N}$, Diffusion Maps maps the points to the leading eigenvectors of the RW-Laplacian matrix of the data as follows:

$$X \to \left( (1 - \lambda_2)^t v_2 \quad \cdots \quad (1 - \lambda_{k+1})^t v_{k+1} \right) = Y,$$

where $X \in \mathbb{R}^{n \times d}$ is a matrix containing each input point as a row, and $Y \in \mathbb{R}^{n \times k}$ is a matrix containing each of the representations as a row. As Sep-SpectralNet is able to approximate both the eigenvectors and eigenvalues of the Laplacian matrix, it is able to make Diffusion Maps generalizable and efficient (Sec. 5.1).

### 4.5 Evaluating UMAP embedding - Grassmann Score

Common evaluation methods for dimensionality reduction, particularly for visualization, are predominantly focused on local structures. For instance, McInnes et al. (2018); Kawase et al. (2022) use kNN accuracy and Trustworthiness, which only account for the local neighborhoods of each point while overlooking global structures such as cluster separation. One global evaluation method is the Silhouette score, which measures the clustering quality of the classes within the embedding space. However, this score does not capture the preservation of the overall global structure.

To address this gap, we propose a new evaluation method, specifically appropriate for assessing global structure preservation in graph-based dimensionality reduction methods (e.g., UMAP, t-SNE). The leading eigenvectors of the Laplacian matrix are known to encode crucial global information about the graph (Belkin & Niyogi, 2003). Thus, we measure the distance between the global structures of the original and embedding manifolds using the Grassmann distance between the first eigenvectors of their respective Laplacian matrices. We refer to this method as the *Grassmann Score* (GS).

It is important to note that GS includes a hyper-parameter - the number of eigenvectors considered. Increasing the number of eigenvectors incorporates more local structure into the evaluation. A natural choice for this hyperparameter is 2, which corresponds to comparing the Fiedler vectors (i.e., the second eigenvectors of the Laplacian). The Fiedler vector is well known for encapsulating the global information of a graph (Fiedler, 1973; 1975). Accordingly, for simplicity and unless specified otherwise, the GS is computed using the first two eigenvectors. Fig. 4 demonstrates GS (alongside Silhouette and kNN scores for comparison) on a few embeddings of two tangent spheres, independently to the embedding methods. Notably, the embedding on the right appears to best preserve the global structure, as indicated by the smallest GS value. In contrast, the kNN scores are comparable across all embeddings (e.g., kNN ignores separation of an existing class), and the Silhouette score even favors other embeddings. In App. D we mathematically formalize GS and provide additional examples of embeddings and their corresponding GS. These examples further support the intuition that GS effectively captures global structure preservation better than previous measures.

## 5 Experiments

### 5.1 Eigenvectors Separation - Generalizable SE

In this section, we demonstrate Sep-SpectralNet's ability to approximate and generalize the SE using four real-world datasets: CIFAR10 (via their CLIP embedding); Appliances Energy Prediction dataset (Candanedo, 2017); Kuzushiji-MNIST (KMNIST) dataset (Clanuwat et al., 2018); Parkinsons Telemonitoring dataset (Tsanas & Little, 2009). Particularly, we compare our results with SpectralNet, which has been empirically shown to approximate the SE space. However, as our results demonstrate, SpectralNet is insufficient for accurately approximating SE. For additional technical details regarding the datasets, architectures and training procedures, we refer the reader to Appendix G.

**Evaluation Metrics.** To assess the approximation of each eigenvector (i.e., the SE), we compute the $\sin^2$ of the angle between each predicted and ground truth vector. This can be viewed as the 1-dimensional case of the Grassmann distance, a well-known metric for comparing equidimensional linear subspaces (see

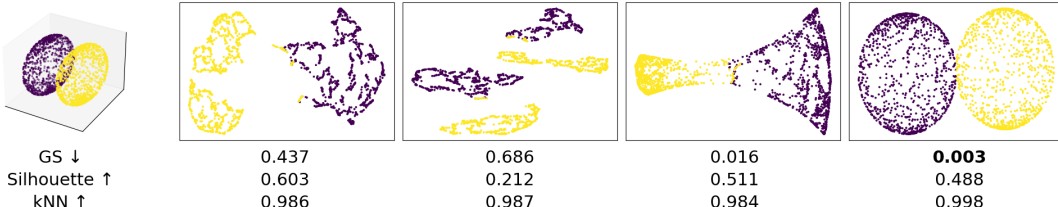

| | | | | |
|---|---|---|---|---|
| GS ↓ | 0.437 | 0.686 | 0.016 | **0.003** |
| Silhouette ↑ | 0.603 | 0.212 | 0.511 | 0.488 |
| kNN ↑ | 0.986 | 0.987 | 0.984 | 0.998 |

Figure 4: **Grassmann Score (GS) reflects global structure preservation.** A demonstration of the alignment between intuitive expectations and GS on a toy dataset of two 3D tangent spheres. Four possible 2D embeddings of the dataset are shown, alongside their corresponding GS, kNN accuracy, and Silhouette score. The rightmost visualization best preserves the global structure, forming two tangent circles in 2D. Notably, kNN fails to distinguish between the embeddings, while Silhouette favors the most clustered one, despite its poor reflection of the original structure. In contrast, GS successfully captures the preservation of global structure, assigning the highest score to the most faithful embedding.

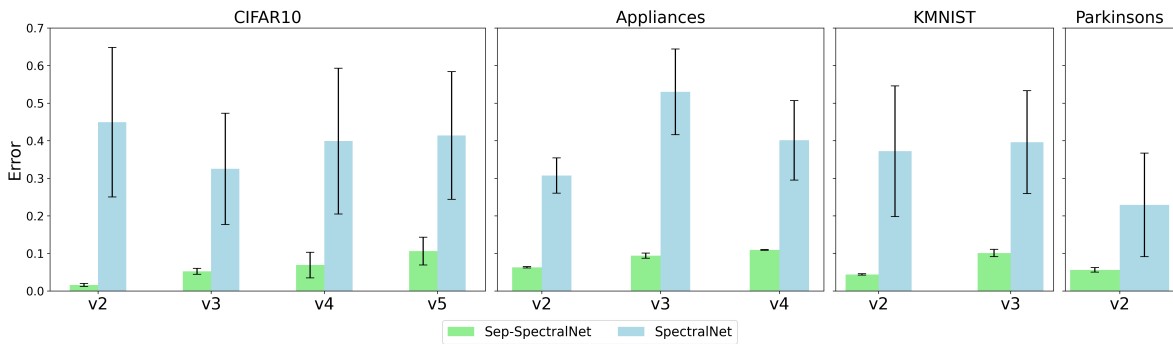

Figure 5: **Sep-SpectralNet successfully separates the eigenvectors.** A comparison between Sep-SpectralNet and SpectralNet of SE and Fiedler Vector (v2) approximation on real-world datasets. The values are the mean and standard deviation of the $\sin^2$ distance between the predicted and true eigenvector of the test set, over 10 runs. Lower is better. Notably, Sep-SpectralNet's errors are significantly smaller than SpectralNet's, supporting Sep-SpectralNet's ability to separate the eigenvectors.

formalization in App. D). Concerning the eigenvalues approximation evaluation, we measure the Pearson Correlation between the predicted and true eigenvalues (computed via SVD).

Fig. 5 presents our results on the real-world datasets. Sep-SpectralNet's output is used directly, while SpectralNet's predicted eigenvectors are resorted to minimize the mean $\sin^2$ distance. The results clearly show that Sep-SpectralNet consistently produces significantly more accurate SE approximations compared to SpectralNet, due to the improved separation of the eigenvectors.

Additionaly, note the Sep-SpectralNet approximates the eigenvalues as well. When concerning a series of Laplacian eigenvalues, the most important property is the relative increase of the eigenvalues (Coifman & Lafon, 2006b). Sep-SpectralNet demonstrates a strong ability to approximate this property. To see this, we repeated Sep-SpectralNet's eigenvalue approximation (10 times) and calculated the Pearson correlation between the predicted and accurate eigenvalues vector. We compared the first 10 eigenvalues. The resulting mean correlation and standard deviation are: Parkinsons Telemonitoring: $\mathbf{0.917}_{\pm 0.0381}$; Appliances Energy Prediction: $\mathbf{0.839}_{\pm 0.0342}$;

## 5.2 Scalability

Noteworthy, Sep-SpectralNet not only generalizes effectively but also does so more quickly than the most scalable (yet non-generalizable) existing methods. Fig. 2b demonstrates this point on the toy moon dataset

- a 2D moon linearly embedded into 10D input space. To evaluate scalability, we measured the computation time required for SE approximation, for an increasing number of samples. We compared the results with the three most popular methods for sparse matrix decomposition, which are currently the fastest implementations: ARPACK (Lehoucq et al., 1998), LOBPCG (Benner & Mach, 2011), and AMG (Brandt et al., 1984). For each number of samples, we calculated the Laplacian matrix that is 99% sparse. Each method was executed five times, initialized with different seeds. Notably, for higher numbers of samples, Sep-SpectralNet converges significantly faster. Notably, Sep-SpectralNetmaintains the same scalability as SpectralNet, since the separation step incurs negligible overhead compared to the overall training time.

### 5.3 NUMAP - generalizable UMAP

In this section, we demonstrate NUMAP's ability to preserve global structure, while enabling fast inference of test points, and it's ability to enable time-series UMAP visualization. We compare our results with P. UMAP, with the target dimensionality set to 2. We refer to App. E for various ablations.

We consider four real-world datasets: CIFAR10 (via their CLIP embedding); Appliances Energy Prediction dataset; Wine (Aeberhard & Forina, 1992); Banknote Authentication (Lohweg, 2012). For additional technical details regarding the datasets, architectures and training procedures, we refer to App. G.

**Evaluation Metrics.** To evaluate and compare the embeddings, we employed both local and global evaluation metrics. For local evaluation, we used the well-established accuracy of a kNN classifier on the embeddings (McInnes et al., 2018; Sainburg et al., 2021), which is applicable only on classed data. For global evaluation, we use GS (see discussion in Sec. 4.5).

**Local Structure.** Tab. 2 presents the kNN results on the real-world datasets. These results are comparable across the methods. Namely, NUMAP does not compromise on local structure.

**Global Structure.** Tab. 3 presents the GS results on these datasets. The results underscore that NUMAP better captures the global structure. Namely, NUMAP enhances global structure preservation while not compromising local structure preservation.

In Fig. 1 and Fig. 6, we supplement the empirical results with qualitative examples. Fig. 1 presents simple non-linear synthetic 3-dimensional structures and their 2-dimensional visualizations using UMAP (non-parametric), P. UMAP and NUMAP. These examples are particularly insightful, as the expected outcome of a good visualization is known. UMAP (using its default configuration, SE initialization) accurately preserves the global structure in its 2-dimensional representations, but lack the ability to generalize to unseen points. Among the generalizable methods (Fig. 1b), P. UMAP fails to preserve the global structure: in the top two rows, it does not separate the clusters, while in the bottom row, it introduces undesired color

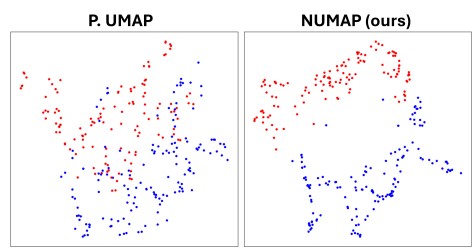

Figure 6: **NUMAP and P. UMAP visualization of the Banknote dataset test set.** NUMAP's enhanced ability to preserve global structure is evident in the clearer separation between clusters compared to P. UMAP.

overlaps. In contrast, NUMAP effectively preserves these separations and avoids unnecessary overlapping. Fig. 6 further demonstrates NUMAP's ability to preserve global structure, as evidenced by the improved class separation in the Banknote dataset.

**Time-series data visualization.** Fig. 7 shows a simulation time-series data, which can be viewed as a simulation of cellular differentiation. Specifically, we may consider differentiation of hematopoietic stem cells (also known as blood stem cells), which are known to differentiate into many types of blood cells, to T-cells. The process involves two kinds of cells (represented by their gene expressions; red and blue samples in the figure). One represents stem cells, while the other T-cells. A group of cells (colored in pink in the figure) then gradually transitions from stem cells to T-cells. At the top row we use UMAP to visualize each time

| Metric | Method | Cifar10 | Appliances | Wine | Banknote |
|---|---|---|---|---|---|
| kNN ↑ | P. UMAP | $0.908_{\pm 0.004}$ | - | $0.953_{\pm 0.033}$ | $0.927_{\pm 0.023}$ |
| | **NUMAP (ours)** | $0.905_{\pm 0.004}$ | - | $0.950_{\pm 0.024}$ | $0.986_{\pm 0.004}$ |

Table 2: **NUMAP local structure preservation is comparable with P. UMAP.** kNN accuracy results of P. UMAP and NUMAP on labeled real-world datasets (Appliances dataset is excluded due to the absence of labels). The values are the mean and standard deviation measures on the test set, over 10 runs.

| Metric | Method | Cifar10 | Appliances | Wine | Banknote |
|---|---|---|---|---|---|
| GS ↓ | P. UMAP | $0.133_{\pm 0.069}$ | $0.710_{\pm 0.293}$ | $0.549_{\pm 0.180}$ | $0.722_{\pm 0.079}$ |
| | **NUMAP (ours)** | $\mathbf{0.054}_{\pm 0.021}$ | $\mathbf{0.261}_{\pm 0.020}$ | $\mathbf{0.429}_{\pm 0.124}$ | $\mathbf{0.618}_{\pm 0.131}$ |
| | p-value | 0.004 | 0.0002 | 0.118 | 0.056 |

Table 3: **NUMAP better preserves global structure.** P. UMAP and NUMAP Grassmann Score (GS) results on real-world datasets. The values are the mean and standard deviation measures on the test set, over 10 runs. The best in mean is highlighted. Notably, NUMAP achieves better global structure preservation.

step, while at the bottom we train NUMAP on the first two time-steps and only inference the rest. UMAP is inconsistent over time-steps, which makes it impractical for understanding change and progression. It also has to train the embeddings at each time-step separately. In contrast, NUMAP only trains on the first two time-steps, and the embeddings of the later time-steps are immediate from inference. This also enables consistency over time, and makes the trend and process visible and understandable.

## 6 Conclusions

We introduced Sep-SpectralNet, a deep-learning approach for approximate SE. Sep-SpectralNet addresses the three primary drawbacks of current SE implementation: generalizability, scalability and eigenvectors separation. By incorporating a post-processing diagonalization step, Sep-SpectralNet enables eigenvectors separation without compromising SpectralNet's generalizability or scalability. Remarkably, this one-shot post-processing operation lays the groundwork for a wide range of new applications of SE, which would not have been possible without its scalable and generalizable implementation. It also presents a promising pathway to enhance current applications of SE.

In particular, we also presented NUMAP, a novel extension of Sep-SpectralNet for generalizable UMAP visualization. We believe the integration of generalizable SE with deep learning can have a significant impact on unsupervised learning methods. However, exploring alternative inputs to the UMAP loss network is an interesting direction for future work. Finally, further research should delve into exploring the applications of SE across various fields.

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

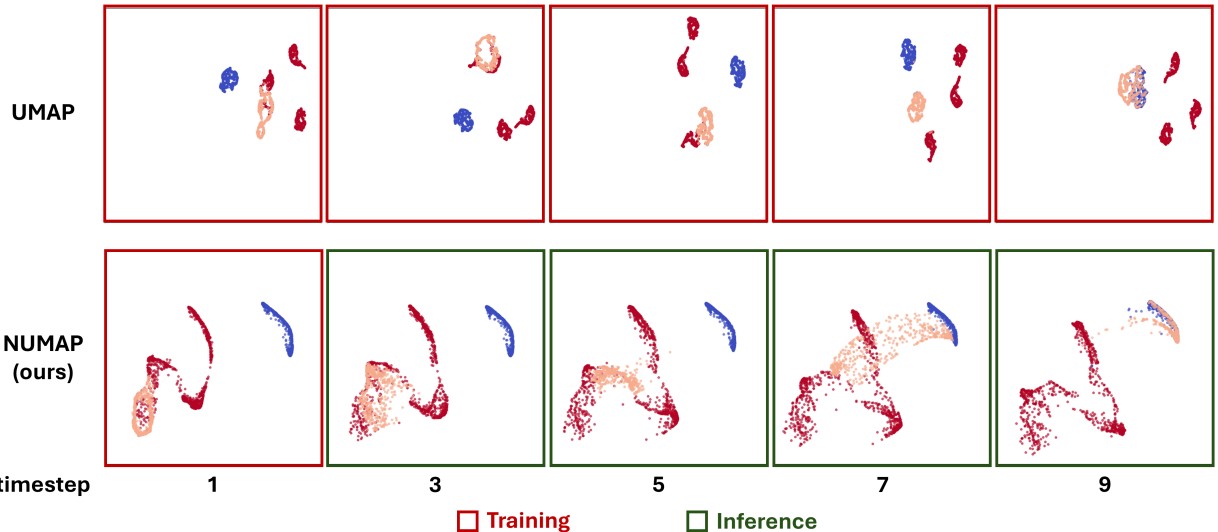

Figure 7: **Time-series data visualization using NUMAP.** A comparison between UMAP and NUMAP visualizations of a dynamical system. In the UMAP visualization, the structure and cluster positions vary significantly across timesteps, hindering the interpretability of the underlying process. In contrast, NUMAP produces consistent structures over time, clearly revealing the transition of the pink points between the red and blue regions. Notably, NUMAP requires training only on the first two timesteps, enabling efficient generalization to the rest of the sequence.

Dominique Beaini, Saro Passaro, Vincent Létourneau, Will Hamilton, Gabriele Corso, and Pietro Liò. Directional graph networks. In International Conference on Machine Learning, pp. 748–758. PMLR, 2021. doi: https://doi.org/10.48550/arXiv.2010.02863.

Mikhail Belkin and Partha Niyogi. Laplacian eigenmaps for dimensionality reduction and data representation. Neural computation, 15(6):1373–1396, 2003. doi: 10.1162/089976603321780317.

Mikhail Belkin and Partha Niyogi. Convergence of laplacian eigenmaps. Advances in neural information processing systems, 19, 2006. doi: https://doi.org/10.7551/mitpress/7503.003.0021.

Peter Benner and Thomas Mach. Locally optimal block preconditioned conjugate gradient method for hierarchical matrices. PAMM, 11(1):741–742, 2011. doi: https://doi.org/10.1002/pamm.201110360.

Achi Brandt, Steve McCormick, and John Ruge. Algebraic multigrid (amg) for sparse matrix equations. Sparsity and its Applications, 257, 1984.

Jiayue Cai, Aiping Liu, Taomian Mi, Saurabh Garg, Wade Trappe, Martin J McKeown, and Z Jane Wang. Dynamic graph theoretical analysis of functional connectivity in parkinson's disease: The importance of fiedler value. IEEE journal of biomedical and health informatics, 23(4):1720–1729, 2018. doi: 10.1109/JBHI.2018.2875456.

Kieran Campbell, Chris P Ponting, and Caleb Webber. Laplacian eigenmaps and principal curves for high resolution pseudotemporal ordering of single-cell rna-seq profiles. bioRxiv, pp. 027219, 2015. doi: https://doi.org/10.1101/027219.

Luis Candanedo. Appliances Energy Prediction. UCI Machine Learning Repository, 2017. DOI: https://doi.org/10.24432/C5VC8G.

Ziyu Chen, Yingzhou Li, and Xiuyuan Cheng. Specnet2: Orthogonalization-free spectral embedding by neural networks. arXiv preprint arXiv:2206.06644, 2022. doi: https://doi.org/10.48550/arXiv.2206.06644.

Eliodoro Chiavazzo, Roberto Covino, Ronald R Coifman, C William Gear, Anastasia S Georgiou, Gerhard Hummer, and Ioannis G Kevrekidis. Intrinsic map dynamics exploration for uncharted effective free-energy landscapes. Proceedings of the National Academy of Sciences, 114(28):E5494–E5503, 2017. doi: https://doi.org/10.1073/pnas.1621481114.

Tarin Clanuwat, Mikel Bober-Irizar, Asanobu Kitamoto, Alex Lamb, Kazuaki Yamamoto, and David Ha. Deep learning for classical japanese literature, 2018.

Ronald R Coifman and Stéphane Lafon. Geometric harmonics: a novel tool for multiscale out-of-sample extension of empirical functions. Applied and Computational Harmonic Analysis, 21(1):31–52, 2006a. doi: https://doi.org/10.1016/j.acha.2005.07.005.

Ronald R Coifman and Stéphane Lafon. Diffusion maps. Applied and computational harmonic analysis, 21 (1):5–30, 2006b. doi: https://doi.org/10.1016/j.acha.2006.04.006.

Sebastian Damrich, Jan Niklas Böhm, Fred A Hamprecht, and Dmitry Kobak. From $t$-sne to umap with contrastive learning. arXiv preprint arXiv:2206.01816, 2022. doi: https://doi.org/10.48550/arXiv.2206.01816.

Michaël Defferrard, Xavier Bresson, and Pierre Vandergheynst. Convolutional neural networks on graphs with fast localized spectral filtering. Advances in neural information processing systems, 29, 2016. doi: https://doi.org/10.48550/arXiv.1606.09375.

Li Deng. The mnist database of handwritten digit images for machine learning research [best of the web]. IEEE Signal Processing Magazine, 29(6):141–142, 2012. doi: 10.1109/MSP.2012.2211477.

Zhijie Deng, Jiaxin Shi, and Jun Zhu. Neuralef: Deconstructing kernels by deep neural networks. In International Conference on Machine Learning, pp. 4976–4992. PMLR, 2022. doi: https://doi.org/10.48550/arXiv.2205.00165.

Debo Dong, Cheng Luo, Xavier Guell, Yulin Wang, Hui He, Mingjun Duan, Simon B Eickhoff, and Dezhong Yao. Compression of cerebellar functional gradients in schizophrenia. Schizophrenia bulletin, 46(5):1282–1295, 2020. doi: 10.1093/schbul/sbaa016.

Klaus Eckelt, Andreas Hinterreiter, Patrick Adelberger, Conny Walchshofer, Vaishali Dhanoa, Christina Humer, Moritz Heckmann, Christian Steinparz, and Marc Streit. Visual exploration of relationships and structure in low-dimensional embeddings. IEEE Transactions on Visualization and Computer Graphics, 29(7):3312–3326, 2023. doi: 10.1109/TVCG.2022.3156760.

Miroslav Fiedler. Algebraic connectivity of graphs. Czechoslovak mathematical journal, 23(2):298–305, 1973. doi: 10.21136/CMJ.1973.101168.

Miroslav Fiedler. A property of eigenvectors of nonnegative symmetric matrices and its application to graph theory. Czechoslovak mathematical journal, 25(4):619–633, 1975. doi: 10.21136/CMJ.1975.101357.

Ian Gemp, Brian McWilliams, Claire Vernade, and Thore Graepel. Eigengame: Pca as a nash equilibrium. arXiv preprint arXiv:2010.00554, 2020. doi: https://doi.org/10.48550/arXiv.2010.00554.

Danial Ghamari, Roberto Covino, and Pietro Faccioli. Sampling a rare protein transition using quantum annealing. Journal of Chemical Theory and Computation, 20(8):3322–3334, 2024. doi: https://doi.org/10.1021/acs.jctc.3c01174.

Aristides Gionis, Piotr Indyk, Rajeev Motwani, et al. Similarity search in high dimensions via hashing. In Vldb, volume 99, pp. 518–529, 1999.

Aldo Glielmo, Brooke E Husic, Alex Rodriguez, Cecilia Clementi, Frank Noé, and Alessandro Laio. Unsupervised learning methods for molecular simulation data. Chemical Reviews, 121(16):9722–9758, 2021. doi: https://doi.org/10.1021/acs.chemrev.0c01195.

Thomas J Hardin, Michael Chandross, Rahul Meena, Spencer Fajardo, Dimitris Giovanis, Ioannis Kevrekidis, Michael L Falk, and Michael D Shields. Revealing the hidden structure of disordered materials by parameterizing their local structural manifold. Nature communications, 15(1):4424, 2024. doi: https://doi.org/10.1038/s41467-024-48449-0.

Ori Katz, Ronen Talmon, Yu-Lun Lo, and Hau-Tieng Wu. Alternating diffusion maps for multimodal data fusion. Information Fusion, 45:346–360, 2019. doi: https://doi.org/10.1016/j.inffus.2018.01.007.

Yoshiaki Kawase, Kosuke Mitarai, and Keisuke Fujii. Parametric t-stochastic neighbor embedding with quantum neural network. Physical Review Research, 4(4):043199, 2022. doi: https://doi.org/10.48550/arXiv.2202.04238.

Paulo Valente Klaine, Muhammad Ali Imran, Oluwakayode Onireti, and Richard Demo Souza. A survey of machine learning techniques applied to self-organizing cellular networks. IEEE Communications Surveys & Tutorials, 19(4):2392–2431, 2017. doi: 10.1109/COMST.2017.2727878.

Dmitry Kobak and George C Linderman. Initialization is critical for preserving global data structure in both t-sne and umap. Nature biotechnology, 39(2):156–157, 2021. doi: https://doi.org/10.1038/nbt.4314.

Sibsankar Kundu, Dan C Sorensen, and George N Phillips Jr. Automatic domain decomposition of proteins by a gaussian network model. Proteins: Structure, Function, and Bioinformatics, 57(4):725–733, 2004.

Stephane Lafon, Yosi Keller, and Ronald R Coifman. Data fusion and multicue data matching by diffusion maps. IEEE Transactions on pattern analysis and machine intelligence, 28(11):1784–1797, 2006. doi: 10.1109/TPAMI.2006.223.

Roy R Lederman and Ronen Talmon. Learning the geometry of common latent variables using alternating-diffusion. Applied and Computational Harmonic Analysis, 44(3):509–536, 2018. doi: https://doi.org/10.1016/j.acha.2015.09.002.

Richard B Lehoucq, Danny C Sorensen, and Chao Yang. ARPACK users' guide: solution of large-scale eigenvalue problems with implicitly restarted Arnoldi methods. SIAM, 1998.

Jersson X Leon-Medina, Núria Parés, Maribel Anaya, Diego A Tibaduiza, and Francesc Pozo. Data classification methodology for electronic noses using uniform manifold approximation and projection and extreme learning machine. Mathematics, 10(1):29, 2021. doi: https://doi.org/10.3390/math10010029.

Ren-Cang Li. Rayleigh quotient based optimization methods for eigenvalue problems. In Matrix Functions and Matrix Equations, pp. 76–108. World Scientific, 2015. doi: https://doi.org/10.1142/9789814675772_0004.

Derek Lim, Joshua Robinson, Lingxiao Zhao, Tess Smidt, Suvrit Sra, Haggai Maron, and Stefanie Jegelka. Sign and basis invariant networks for spectral graph representation learning. arXiv preprint arXiv:2202.13013, 2022. doi: https://doi.org/10.48550/arXiv.2202.13013.

Leland Zhang Liu. Umap-pytorch: Umap (uniform manifold approximation and projection) in pytorch, 2024. URL https://github.com/elyxlz/umap_pytorch.

Volker Lohweg. Banknote Authentication. UCI Machine Learning Repository, 2012. DOI: https://doi.org/10.24432/C55P57.

George Ma, Yifei Wang, and Yisen Wang. Laplacian canonization: A minimalist approach to sign and basis invariant spectral embedding. Advances in Neural Information Processing Systems, 36, 2024. doi: https://doi.org/10.48550/arXiv.2310.18716.

Leland McInnes, John Healy, and James Melville. Umap: Uniform manifold approximation and projection for dimension reduction. arXiv preprint arXiv:1802.03426, 2018. doi: https://doi.org/10.48550/arXiv.1802.03426.

Gal Mishne, Uri Shaham, Alexander Cloninger, and Israel Cohen. Diffusion nets. Applied and Computational Harmonic Analysis, 47(2):259–285, 2019. doi: https://doi.org/10.48550/arXiv.1506.07840.

Evert J Nyström. Über die praktische auflösung von integralgleichungen mit anwendungen auf randwertaufgaben. 1930. doi: https://doi.org/10.1007/BF02547521.

Antonio Ortega, Pascal Frossard, Jelena Kovačević, José MF Moura, and Pierre Vandergheynst. Graph signal processing: Overview, challenges, and applications. Proceedings of the IEEE, 106(5):808–828, 2018. doi: 10.1109/JPROC.2018.2820126.

Bo-yong Park, Seok-Jun Hong, Sofie L Valk, Casey Paquola, Oualid Benkarim, Richard AI Bethlehem, Adriana Di Martino, Michael P Milham, Alessandro Gozzi, BT Thomas Yeo, et al. Differences in subcortico-cortical interactions identified from connectome and microcircuit models in autism. Nature communications, 12(1):2225, 2021. doi: https://doi.org/10.1038/s41467-021-21732-0.

Bo-yong Park, Valeria Kebets, Sara Larivière, Meike D Hettwer, Casey Paquola, Daan van Rooij, Jan Buitelaar, Barbara Franke, Martine Hoogman, Lianne Schmaal, et al. Multiscale neural gradients reflect transdiagnostic effects of major psychiatric conditions on cortical morphology. Communications biology, 5(1):1024, 2022. doi: https://doi.org/10.1038/s42003-022-03963-z.

David Pfau, Stig Petersen, Ashish Agarwal, David GT Barrett, and Kimberly L Stachenfeld. Spectral inference networks: Unifying deep and spectral learning. arXiv preprint arXiv:1806.02215, 2018. doi: https://doi.org/10.48550/arXiv.1806.02215.

Tim Sainburg, Leland McInnes, and Timothy Q Gentner. Parametric umap embeddings for representation and semisupervised learning. Neural Computation, 33(11):2881–2907, 2021. doi: https://doi.org/10.48550/arXiv.2009.12981.

Finn Schofield and Andrew Lensen. Using genetic programming to find functional mappings for umap embeddings. In 2021 IEEE Congress on Evolutionary Computation (CEC), pp. 704–711. IEEE, 2021. doi: 10.1109/CEC45853.2021.9504848.

Uri Shaham, Kelly Stanton, Henri Li, Boaz Nadler, Ronen Basri, and Yuval Kluger. Spectralnet: Spectral clustering using deep neural networks. In Proc. ICLR 2018, 2018. doi: https://doi.org/10.48550/arXiv.1801.01587.

SJ Shepherd, CB Beggs, and Sue Jones. Amino acid partitioning using a fiedler vector model. European Biophysics Journal, 37:105–109, 2007. doi: https://doi.org/10.1007/s00249-007-0182-y.

Zuoqiang Shi. Convergence of laplacian spectra from random samples. arXiv preprint arXiv:1507.00151, 2015. doi: https://doi.org/10.48550/arXiv.1507.00151.

Or Streicher, Ido Cohen, and Guy Gilboa. Basis: batch aligned spectral embedding space. In Proceedings of the IEEE/CVF Conference on Computer Vision and Pattern Recognition, pp. 10396–10405, 2023. doi: https://doi.org/10.48550/arXiv.2211.16960.

Edric Tam and David Dunson. Fiedler regularization: Learning neural networks with graph sparsity. In International Conference on Machine Learning, pp. 9346–9355. PMLR, 2020. doi: https://doi.org/10.48550/arXiv.2003.00992.

Athanasios Tsanas and Max Little. Parkinsons Telemonitoring. UCI Machine Learning Repository, 2009. DOI: https://doi.org/10.24432/C5ZS3N.

Laurens Van Der Maaten. Learning a parametric embedding by preserving local structure. In Artificial intelligence and statistics, pp. 384–391. PMLR, 2009.

Ulrike Von Luxburg. A tutorial on spectral clustering. Statistics and computing, 17:395–416, 2007. doi: https://doi.org/10.48550/arXiv.0711.0189.

Han Xiao, Kashif Rasul, and Roland Vollgraf. Fashion-mnist: a novel image dataset for benchmarking machine learning algorithms. arXiv preprint arXiv:1708.07747, 2017. doi: 10.48550/arXiv.1708.07747.

Yuxuan Richard Xie, Daniel C Castro, Stanislav S Rubakhin, Timothy J Trinklein, Jonathan V Sweedler, and Fan Lam. Integrative multiscale biochemical mapping of the brain via deep-learning-enhanced high-throughput mass spectrometry. Nature Methods, 21:521–530, 2024. doi: https://doi.org/10.1038/s41592-024-02171-3.

Bingxian Xu and Guangzheng Zhang. Robust parametric umap for the analysis of single-cell data. bioRxiv, pp. 2023–11, 2023. doi: https://doi.org/10.1101/2023.11.14.567092.

Eunkyung Yoo, Hyeonseop Song, Taehyeong Kim, and Chul Lee. Online learning of open-set speaker identification by active user-registration. In INTERSPEECH, pp. 5065–5069, 2022.

Ziwei Zhang, Peng Cui, Jian Pei, Xin Wang, and Wenwu Zhu. Eigen-gnn: A graph structure preserving plug-in for gnns. IEEE Transactions on Knowledge and Data Engineering, 35(3):2544–2555, 2021. doi: https://doi.org/10.48550/arXiv.2006.04330.

Wenwei Zheng, Bo Qi, Mary A Rohrdanz, Amedeo Caflisch, Aaron R Dinner, and Cecilia Clementi. Delineation of folding pathways of a $\beta$-sheet miniprotein. The Journal of Physical Chemistry B, 115(44): 13065–13074, 2011. doi: https://doi.org/10.1021/jp2076935.

Wenwei Zheng, Mary A Rohrdanz, and Cecilia Clementi. Rapid exploration of configuration space with diffusion-map-directed molecular dynamics. The journal of physical chemistry B, 117(42):12769–12776, 2013. doi: https://doi.org/10.1021/jp401911h.

Qiyao Zhu and Tamar Schlick. A fiedler vector scoring approach for novel rna motif selection. The Journal of Physical Chemistry B, 125(4):1144–1155, 2021. doi: https://doi.org/10.1021/acs.jpcb.0c10685.

## A  Proof of Lemma 1

First, we remind an important property of the Rayleigh Quotient.

**Remark 1.** *The Rayleigh Quotient of a positive semi-definite matrix $L \in \mathbb{R}^{n \times n}$ with eigenvectors $v_1, \ldots, v_n$ corresponding to the eigenvalues $\lambda_1 \leq \cdots \leq \lambda_n$, $R_L$ satisfies $arg\min_{||v||=1} R_L(v) = v_1$ and for each $i > 1$ $arg\min_{||v||=1} R_L(v) = v_i$ for $v \perp v_1, \ldots, v_{i-1}$ (Li, 2015).*

**Lemma 1.** Let $L \in \mathbb{R}^{n \times n}$ be an Unnormalized Laplacian matrix and $R_L : O(n, k) \to \mathbb{R}$ its corresponding RQ, and Let $A$ be a minimizer of $R_L$. Denote $V \in \mathbb{R}^{n \times k}$ as the matrix containing the first $k$ eigenvectors of $L$ as its columns, and $\Lambda$ the corresponding diagonal eigenvalues matrix. Then, there exists an orthogonal matrix $Q \in \mathbb{R}^{k \times k}$ such that $A = VQ$.

*Proof.* As $V$ minimizes $R_L$, we get that $\min_U R_L(U) = R_L(V) = \sum_{i=1}^{k} \lambda_i$, where $0 = \lambda_1 \leq \lambda_2 \leq \cdots \leq \lambda_n$ are the eigenvalues of $L$. This yields

$$R_L(A) = \text{Tr}(A^T LA) = \sum_{i=1}^{k} \lambda_i$$

$A^T LA$ is symmetric, and hence orthogonally diagonalizable, which means there exists an orthogonal matrix $Q \in \mathbb{R}^{k \times k}$ and a diagonal matrix $D \in \mathbb{R}^{k \times k}$ s.t.

$$A^T LA = Q^T DQ$$

Which can be written as

$$(AQ^T)^T L(AQ^T) = D$$

Denoting by $d_1, \ldots, d_k$ the diagonal values of $D$, the last equation yields

$$\sum_{i=1}^{k} d_i = R_L(AQ^T) = R_L(A) = \sum_{i=1}^{k} \lambda_i$$

Note that based on Remark 1 $\lambda_i \leq d_i$ for each $i$, as $AQ^T \in O(n,k)$. Hence, $d_i = \lambda_i$, i.e.,

$$(AQ^T)^T L(AQ^T) = \Lambda$$

As the eigendecomposition of a matrix is unique, this yields $AQ^T = V$, which means $A = VQ$. $\qquad\square$

## B  Algorithm Layouts

---
**Algorithm 1:** SpectralNet training (Shaham et al., 2018)

---
**Input:** $\mathcal{X} \subseteq \mathbb{R}^d$, number of dimensions $k$, batch size $m$
**Output:** Trained $F_\theta$ which approximates the first $k+1$ eigenfunctions up to isometry
1 Randomly initialize the network weights $\theta$
2 **while** $\underline{\mathcal{L}(\theta) \text{ not converged}}$ **do**
3    **Orthogonalization step:**
4    Sample a random minibatch $X$ of size $m$
5    Forward propagate $X$ and compute inputs to orthogonalization layer $\tilde{Y}$
6    Compute the $QR$ factorization $QR = \tilde{Y}$
7    Set the weights of the orthogonalization layer to be $\sqrt{m}R^{-1}$
8    **Gradient step:**
9    Sample a random minibatch $x_1, \ldots, x_m$
10    Compute the $m \times m$ affinity matrix $W$
11    Forward propagate $x_1, \ldots, x_m$ to get $y_1, \ldots, y_m$
12    Compute the loss $\mathcal{L}(\theta)$(Sec. 3.2)
13    Use the gradient of $\mathcal{L}(\theta)$ to tune all $F_\theta$ weights, except those of the output layer;

---

---
**Algorithm 2:** Eigenvectors separation

---
**Input:** $\mathcal{X} \subseteq \mathbb{R}^d$, batch size $m$, Trained $F_\theta$ which approximates the first $k+1$ eigenfunctions up to isometry
**Output:** $F_\theta$ which approximates the leading eigenfunctions
1 T $\leftarrow \lfloor \frac{|\mathcal{X}|}{m} \rfloor$
2 sample T minibatches $X_i \in \mathbb{R}^{m \times d}$
3 Forward propogate all $X_i$ and obtain $F_\theta$ outputs $Y_i \in \mathbb{R}^{m \times k+1}$
4 Compute the $m \times m$ affinity matrices $W_i$
5 compute all corresponding RW-Laplacians $L_i$
6 $\tilde{\Lambda} \leftarrow \frac{1}{T} \sum_i Y_i^T L_i Y_i$
7 Diagonalize $\tilde{\Lambda}$ to get $\tilde{Q}^T$ and the leading eigenvalues
8 Sort the leading eigenvalues, and the columns of $\tilde{Q}^T$ correspondingly
9 $Q^T \leftarrow$ last $k$ columns of $\tilde{Q}^T$
10 To obtain the representation of a new test sample $x_i$, compute $y_i = F_\theta(x_i)Q^T$

---

## C  Implementation's Additional Considerations

### C.1  Time and Space Complexity

Specifying the exact complexity of the method is difficult, As this is a non-convex optimization problem, However, we can discuss the following approximate complexity analysis. Assuming constant input and output

dimensions and a given network architecture, we can take a general view on the complexity of each iteration by the batch size $m$. The heaviest computational operations at each iteration are the nearest-neighbors search, the QR decomposition and the loss computation (i.e., computation of the Rayleigh Quotient). For the nearest-neighbor search, we can use approximation techniques (e.g, LSH Gionis et al. (1999)) which work in almost linear complexity by $m$. A naive implementation of the QR decomposition would lead to an $\mathcal{O}(m^2)$ time complexity. The loss computation also takes $\mathcal{O}(m^2)$ due to the required matrix multiplication. Thereby, the complexity of each iteration is quadratic by the batch size. This is comparable to other approximation techniques such as LOBPCG Benner & Mach (2011) (which also utilizes sparse matrix operations techniques for faster implementation). However, Sep-SpectralNet leverages stochastic training, allowing each iteration to consider only a batch of the data, rather than the entire dataset.

Assessing the complexity of each epoch is now straightforward, and results a time complexity of $\mathcal{O}(nm)$, where $n$, the number of samples, satisfies $n \gg m$. This indicates an almost-linear complexity.

### C.2 Graph Construction

To best capture the structure of the input manifold $\mathcal{D}$, given by a finite number of samples $\mathcal{X}$, we use a similar graph construction method used by Gomez et al. in UMAP (McInnes et al., 2018), proven to capture the local topology of the manifold at each point. However, as opposed to the method in (McInnes et al., 2018), Sep-SpectralNet does not compute the graph of all points, which can lead to scalability hurdles and impose significant memory demands. Instead, Sep-SpectralNet either computes small graphs on each batch, or can be provided by the user with an affinity matrix $W$ corresponding to $\mathcal{X}$. Our practical construction of the graph affinity matrix $W$ is as follows:

Given a distance measure $\delta$ between points, we first compute the $k$-nearest neighbors of each point $x_i$ under $\delta$, $\{x_{i_1}, \ldots, x_{i_k}\}$, and denote

$$\rho_i = \min_j \delta(x_i, x_{i_j}), \ \sigma_i = \text{median}\{\delta(x_i, x_{i_j}) | 1 \leq j \leq k\}$$

Second, we compute the affinity matrix using the Laplace kernel

$$W_{ij} = \begin{cases} \exp\left(\frac{\rho_i - \delta(x_i, x_j)}{\sigma_i}\right) & x_j \in \{x_{i_1}, \ldots, x_{i_k}\} \\ 0 & \text{otherwise} \end{cases}$$

Third, we symmetries $W$ simply by taking $\frac{W + W^T}{2}$.

We refer the reader to McInnes et al. (2018) for further discussion about the graph construction.

## D Grassmann Score

In this section, we provide the formulation for the Grassmann Score (GS) evaluation method, and present simple examples to visualize its meaning.

### D.1 Formalization of GS

Grassmann distance (see Def. 1) is a metric function between equidimensional linear subspaces, where each is represented by an orthogonal matrix containing the basis as its columns. In other words, this is a metric which is invariant under multiplication by an orthogonal matrix.

**Definition 1.** *Given two orthogonal matrices $A, B \in \mathbb{R}^{n \times k}$, the Grassmann Distance between them is defined as:*

$$d_{Gr}(A, B) = \sum_{i=1}^{k} sin^2 \theta_i$$

*where $\theta_i = \arccos \sigma_i(A^T B)$ is the ith principal angle between $A$ and $B$, and $\sigma_i$ is the ith smallest singular value of $A^T B$.*

Assuming we are given a dataset $\mathcal{X} = \{x_1, \ldots, x_n\} \subseteq \mathbb{R}^d$ and a corresponding low-dimensional representation $\mathcal{Y} = \{y_1, \ldots, y_n\} \subseteq \mathbb{R}^k$. We want to evaluate the dissimilarity between the *global structures* of $\mathcal{X}$ and $\mathcal{Y}$. We build graphs from $\mathcal{X}$ and $\mathcal{Y}$, saved as affinity matrices $W_{\mathcal{X}}$ and $W_{\mathcal{Y}}$, respectively. We construct the corresponding Unnormalized Laplacians (see Sec. 3.1) $L_{\mathcal{X}}$ and $L_{\mathcal{Y}}$. We define the matrices $V_{\mathcal{X}}, V_{\mathcal{Y}} \in \mathbb{R}^{n \times t}$ so that their columns are the first $t$ eigenvectors of $L_{\mathcal{X}}, L_{\mathcal{Y}}$, respectively.

Finally, we define the GS of $\mathcal{Y}$ (w.r.t $\mathcal{X}$) as follows:

**Definition 2.** $GS_{\mathcal{X}}(\mathcal{Y}) = d_{Gr}(V_{\mathcal{X}}, V_{\mathcal{Y}})$

$t$ is a hyper-parameter of GS. A reasonable choice would be to take $t = 2$, which is equivalent to measure the Grassmann distance between the Fiedler vectors of the Laplacians. The Fiedler vector is known for its hold of the most important global properties. The larger $t$, the more complicated structures are taken into consideration in the GS computation (which is not necceray desired).

Note that for the construction of the affinity matrices $W_{\mathcal{X}}, W_{\mathcal{Y}}$ we use the same construction scheme detailed in App. C.2. This construction method is similar to the one presented by McInnes et al. (2018), and proved to capture the local topology of the underlying manifold.

It is important to note that GS might ignore the local structures, while concentrating on the global structures (especially for smaller values of $t$). The ultimate goal in visualization is to find a balance between the global and local structure.

## D.2    Additional GS examples

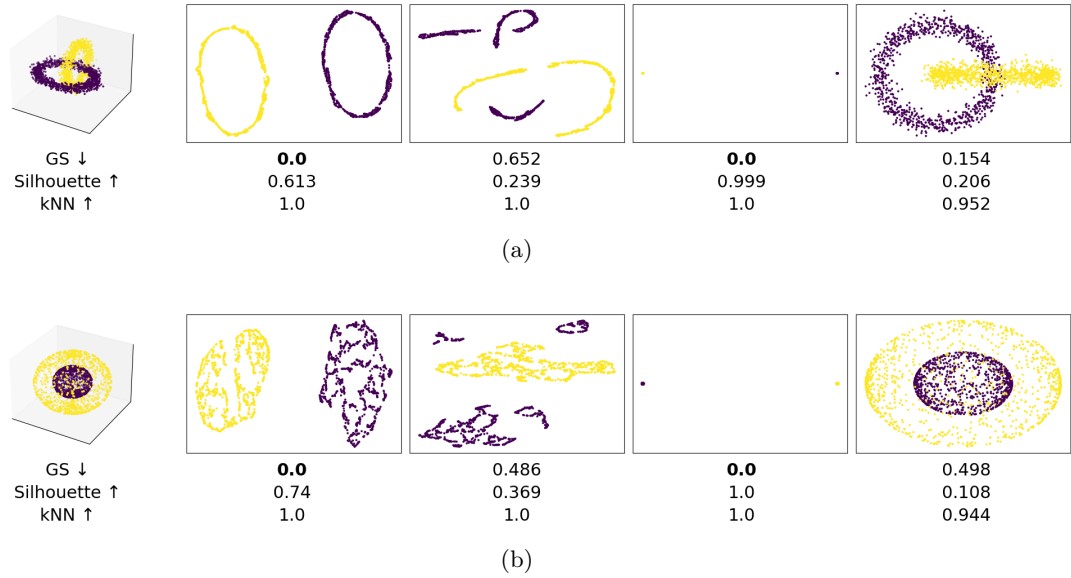

Figure 8: Additional demonstrations of the alignment between the intuitive expectation and the GS results on two toy dataset. Four possible 2-dimensional embeddings of these dataset are provided, along with their corresponding GS, kNN accuracy and Silhouette score. Unlike kNN and Silhouette, GS effectively captures the preservation of global structure.

Fig. 8 depicts two additional demonstrations of the alignment between the intuitive expectation and the GS results on two toy dataset. The basic global structure of both of these datasets is two distinct clusters. This structure is indeed captured by GS. However, kNN gives perfect score also when the one of the clusters is separated. Silhouette score favourites the 2-points embedding. Namely, it trade-offs local structure (i.e., giving lower score for preserving local structure, even when the global properties are the same).

# E    NUMAP Ablation Study

In this section, we start by considering two ablations of NUMAP, which for convenience we name NUMAP-SN and NUMAP-FT. NUMAP-SN is a replication of NUMAP's pipeline with SpectralNet replacing Sep-SpectralNet. That is, we show that eigenvector separation is necessary for NUMAP.

As for NUMAP-FT, this name refers to another method to get a generalizable version of UMAP by an extension of Sep-SpectralNet by fine-tuning the network with UMAP loss. We tried that idea, but were forced to stop this direction, as we stumbled upon the well-known catastrophic forgetting case.

Tab. 4 extends Tab. 2 with the results of NUMAP-SN and NUMAP-FT. Notably, NUMAP-SN local structure preservation are comparable with NUMAP and P. UMAP, as expected, while NUMAP-FT consistently fails in local structure preservation.

Tab. 5 extends Tab. 3 with the results of NUMAP-SN and NUMAP-FT. Notably, NUMAP-SN is less consistent than NUMAP in global structure preservation, resulting in worse performance on the Cifar10 and Banknote datasets. NUMAP-FT fails in global structure preservation on the Cifar10 and Appliances datasets. Although it achieves better GS on the Wine dataset, and comparable global structure preservation on the Banknote dataset, it compromises a lot on local structure preservation (see Tab. 4).

Figure 9 presents an experiment on the simple 2circles dataset. Each row is represented the same experiment, run with a different seed. We trained Sep-SpectralNet to output the 2D SE of the 2circles dataset, as shown in the left column. Then, we initialized a new network, with the same architecture, with the pre-trained weights from Sep-SpectralNet. This network was trained with UMAP loss, as in (Sainburg et al., 2021). We tried different learning-rates for fine-tuning, to best match the desired UMAP embedding (i.e. retaining the local structure), without losing the global structure (e.g., separation of the two clusters). Unfortunately, there was no learning-rate that matched our goals.

| Metric | Method | Cifar10 | Appliances | Wine | Banknote |
|--------|--------|---------|------------|------|----------|
| kNN ↑ | P. UMAP | $0.908_{\pm 0.004}$ | - | $0.953_{\pm 0.033}$ | $0.927_{\pm 0.023}$ |
| | NUMAP (ours) | $0.905_{\pm 0.004}$ | - | $0.950_{\pm 0.024}$ | $0.986_{\pm 0.004}$ |
| | NUMAP-SN | $0.903_{\pm 0.002}$ | - | $0.956_{\pm 0.028}$ | $0.963_{\pm 0.032}$ |
| | **NUMAP-FT** | $0.577_{\pm 0.081}$ | - | $0.364_{\pm 0.053}$ | $0.686_{\pm 0.046}$ |

Table 4: **Local structure preservation ablation.** An extension of Tab. 2 with NUMAP-SN and NUMAP-FT.

| Metric | Method | Cifar10 | Appliances | Wine | Banknote |
|--------|--------|---------|------------|------|----------|
| GS ↓ | P. UMAP | $0.133_{\pm 0.069}$ | $0.710_{\pm 0.293}$ | $0.549_{\pm 0.180}$ | $0.722_{\pm 0.079}$ |
| | NUMAP (ours) | $0.054_{\pm 0.021}$ | $0.261_{\pm 0.020}$ | $0.429_{\pm 0.124}$ | $0.618_{\pm 0.131}$ |
| | NUMAP-SN | $0.281_{\pm 0.357}$ | $0.262_{\pm 0.038}$ | $0.441_{\pm 0.182}$ | $0.780_{\pm 0.197}$ |
| | NUMAP-FT | $0.680_{\pm 0.334}$ | $0.348_{\pm 0.292}$ | $0.002_{\pm 0.001}$ | $0.635_{\pm 0.141}$ |

Table 5: **Global structure preservation ablation.** An extension of Tab. 3 with NUMAP-SN and NUMAP-FT.

Another important question is the necessity of the residual connections. For that, we conduct another ablation study on the Cifar10 dataset. The results are shown in Tab. 6 and Tab. 7. Notably, while the skip connections does not improve local structure preservation, they are crucial for global structure preservation, indicated by the lower GS.

| Metric | Method | Cifar10 |
|--------|--------|---------|
| kNN ↑ | NUMAP (no residual connections) | $0.907_{\pm 0.003}$ |
| | NUMAP | $0.905_{\pm 0.004}$ |

Table 6: Ablation on residual connections for local structure preservation shows comparable performance with and without their use.

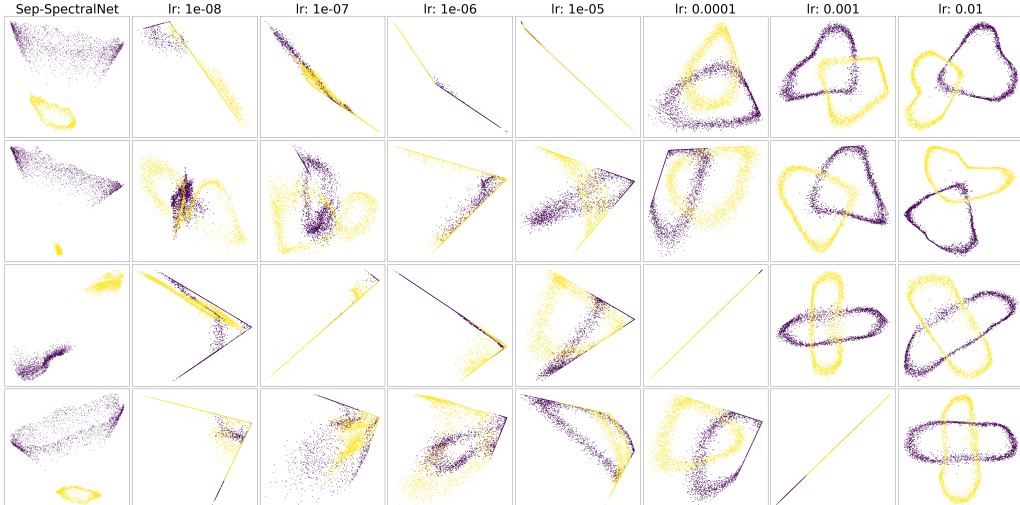

Figure 9: The catastrophic forgetting phenomenon when fine-tuning Sep-SpectralNet to match UMAP performance on the 2circles dataset. Each column represents a fine-tuning using a different learning-rate. Each row is a repetition, initialized with a different seed.

| Metric | Method | Cifar10 |
|--------|--------|---------|
| GS ↓ | NUMAP (no residual connections) | $0.191_{\pm 0.293}$ |
|  | NUMAP | $\mathbf{0.054}_{\pm 0.021}$ |

Table 7: Ablation on residual connections for global structure preservation shows improved performance when they are used.

# F  Additional results

The full results of Fig. 5 are summarized in Tab. 8. The visualizations corresponding to Tab. 3 are depicted in Fig. 10. Fig. 11 extends Fig. 7 with P. UMAP.

Tab. 9 extends Tab. 4 with additional two datasets: MNIST (Deng, 2012) and FashionMNIST (Xiao et al., 2017). Tab. 10 correspondingly extends Tab. 5.

# G  Technical Details

To compute the ground truth SE on the train set and its corresponding eigenvalues, we constructed an affinity matrix $W$ from the train set (as detailed in Appendix C.2), with a number of neighbors detailed in Table 12. After constructing $W$, we computed the leading $k$ eigenvectors of its corresponding Unnormalized Laplacian $L = D - W$ via Python's Numpy SVD or SciPy LOBPCG SVD (depending on the size). To get the generalization ground truth, we constructed an affinity matrix $W$ from the train and test sets combined, computed the leading $k$ eigenvectors of its corresponding Unnormalized Laplacian $L = D - W$, and extracted the representations corresponding to the test samples. We used a train-test split of 80-20 for all datasets.

For the SE implementation via sparse matrix decomposition techniques, we used Python's sklearn.manifold.SpectralEmbedding, using a default configuration (in particular, 10 jobs, 1% neighbors).

The architectures of Sep-SpectralNet's and SpectralNet's networks in all of the experiments were as follows: size = 256; ReLU, size = 256; ReLU, size = 512; ReLU, size = $k+1$; orthonorm. NUMAP's second NN and PUMAP's NN architectures for all datasets was: size = 200; ReLU, size = 200; ReLU, size = 200; ReLU, size = 2; The SE dimensions for NUMAP were: Cifar10 - 10; Appliances - 5; Wine - 10; Banknote - 3; Mnist - 10, FashionMnist - 10. For the datasets in Fig. 1, from top to bottom: Circles - 5, Cylinders - 11, Line - 2.

Table 8: A comparison between Sep-SpectralNet and SpectralNet dimensional SE and Fiedler Vector (FV) approximation on real-world datasets. The values are the mean and standard deviation of the $\sin^2$ distance between the predicted and true eigenvector, over 10 runs. Lower is better. Sep-SpectralNet ability to separate the eigenvectors is evident.

| Dataset | Method | $v_2$ | $v_3$ | $v_4$ | $v_5$ |
|---|---|---|---|---|---|
| Cifar10 | Sep-SpectralNet | $0.016_{\pm0.004}$ | $0.052_{\pm0.008}$ | $0.069_{\pm0.034}$ | $0.106_{\pm0.037}$ |
|  | SpectralNet | $0.449_{\pm0.199}$ | $0.325_{\pm0.148}$ | $0.399_{\pm0.194}$ | $0.414_{\pm0.17}$ |
| Appliances | Sep-SpectralNet | $0.063_{\pm0.002}$ | $0.094_{\pm0.007}$ | $0.109_{\pm0.001}$ | - |
|  | SpectralNet | $0.307_{\pm0.047}$ | $0.530_{\pm0.114}$ | $0.401_{\pm0.106}$ | - |
| KMNIST | Sep-SpectralNet | $0.0044_{\pm0.002}$ | $0.101_{\pm0.010}$ | - | - |
|  | SpectralNet | $0.372_{\pm0.174}$ | $0.396_{\pm0.137}$ | - | - |
| Parkinsons | Sep-SpectralNet | $0.056_{\pm0.006}$ | - | - | - |
|  | SpectralNet | $0.229_{\pm0.138}$ | - | - | - |

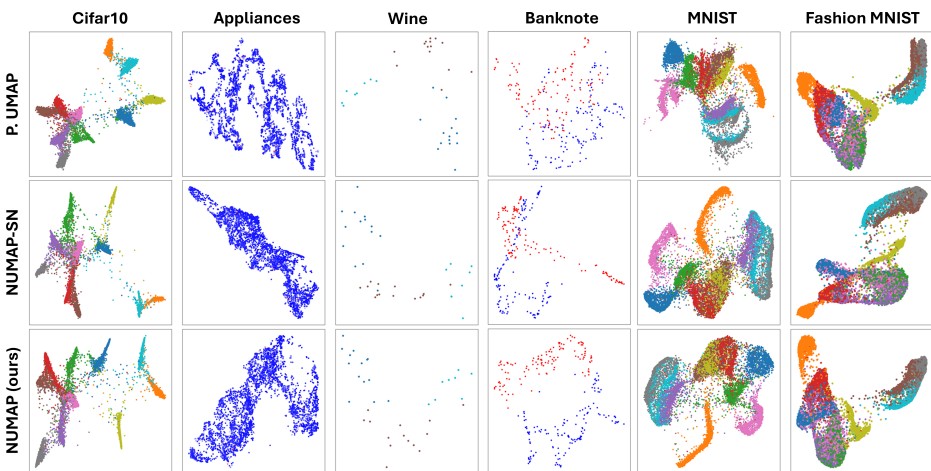

Figure 10: Test set Visualizations of NUMAP, NUMAP-SN and P. UMAP on the datasets corresponding to Tab. 2 and Tab. 3.

The learning rate policy for Sep-SpectralNet and SpectralNet is determined by monitoring the loss on a validation set (a random subset of the training set); once the validation loss did not improve for a specified number of epochs, we divided the learning rate by 10. Training stopped once the learning rate reached $10^{-7}$. In particular, we used the following approximation to determine the patience epochs, where $n$ is the number of samples and $m$ is the batch size: if $\frac{n}{m} \le 25$, we chose the patience to be 10; otherwise, the patience decreases as $\max(1, \frac{250m}{n})$ (i.e., the number of iterations is the deciding feature).

To run UMAP, we used Python's umap-learn implementation (UMAP's formal implementation). We used the built-in initialization option "spectral" (i.e., SE), and initialized contumely with PCA (implemented via Python's sklearn.decomposition.PCA) and Sep-SpectralNet. For Parametric UMAP we used the Pytorch implementaion (Liu, 2024). For all methods we used a default choice of 10 neighbors.

| Metric | Method | Cifar10 | Appliances | Wine | Banknote | Mnist | FashionMnist |
|---|---|---|---|---|---|---|---|
| kNN ↑ | P. UMAP | $0.908_{\pm0.004}$ | - | $0.953_{\pm0.033}$ | $0.927_{\pm0.023}$ | $0.801_{\pm0.010}$ | $0.717_{\pm0.006}$ |
|  | NUMAP (ours) | $0.905_{\pm0.004}$ | - | $0.950_{\pm0.024}$ | $0.986_{\pm0.004}$ | $0.758_{\pm0.009}$ | $0.695_{\pm0.003}$ |
|  | NUMAP-SN | $0.903_{\pm0.002}$ | - | $0.956_{\pm0.028}$ | $0.963_{\pm0.032}$ | $0.750_{\pm0.010}$ | $0.695_{\pm0.006}$ |
|  | NUMAP-FT | $0.577_{\pm0.081}$ | - | $0.364_{\pm0.053}$ | $0.686_{\pm0.046}$ | $0.329_{\pm0.103}$ | $0.153_{\pm0.035}$ |

Table 9: An extension of Tab. 4 with the MNIST and FashionMNIST datasets. Local preservation results are corresponding to Tab. 4. Namely, NUMAP, P. UMAP and NUMAP-SN are comparable, with NUMAP-FT failing to preserve local structure.

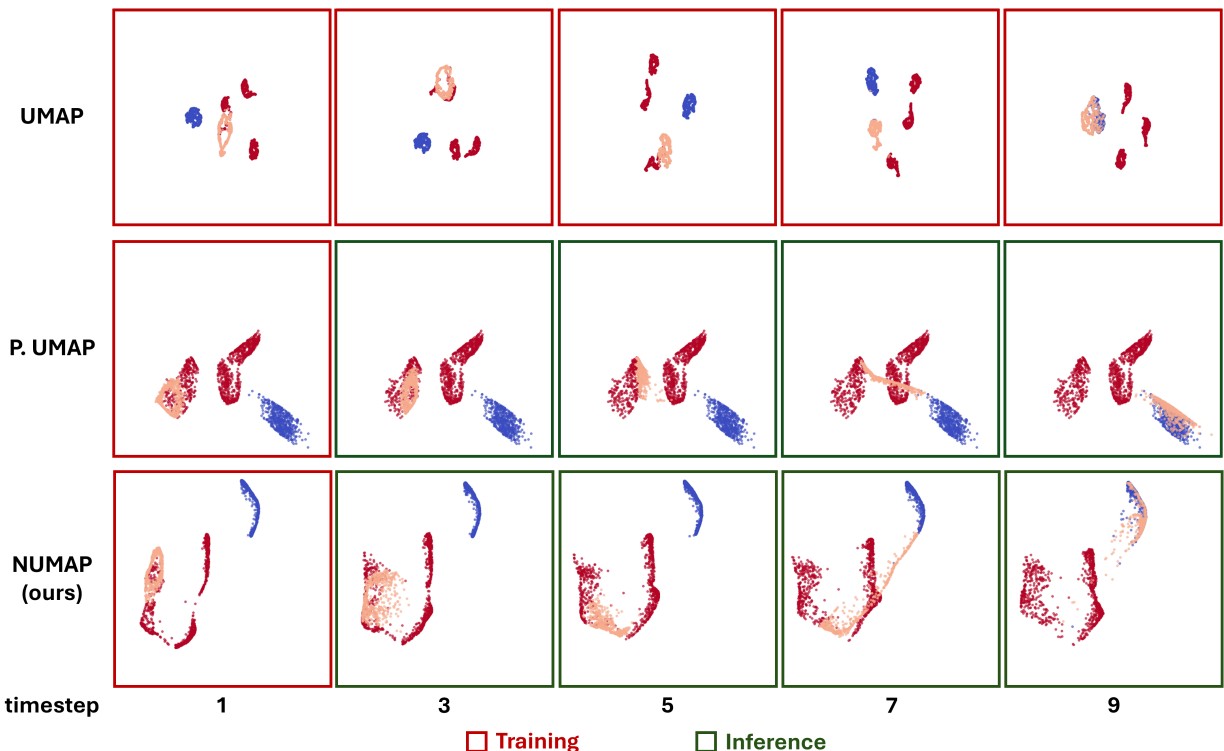

Figure 11: Fig. 7 extension with P. UMAP.

| Metric | Method | Cifar10 | Appliances | Wine | Banknote | Mnist | FashionMnist |
|--------|--------|---------|------------|------|----------|-------|--------------|
| | P. UMAP | $0.133_{\pm 0.069}$ | $0.710_{\pm 0.293}$ | $0.549_{\pm 0.180}$ | $0.722_{\pm 0.079}$ | $0.311_{\pm 0.059}$ | $0.029_{\pm 0.006}$ |
| GS $\downarrow$ | NUMAP (ours) | $0.054_{\pm 0.021}$ | $0.261_{\pm 0.020}$ | $0.429_{\pm 0.124}$ | $0.618_{\pm 0.131}$ | $0.304_{\pm 0.042}$ | $0.032_{\pm 0.022}$ |
| | NUMAP-SN | $0.281_{\pm 0.357}$ | $0.262_{\pm 0.038}$ | $0.441_{\pm 0.182}$ | $0.780_{\pm 0.197}$ | $0.405_{\pm 0.199}$ | $0.036_{\pm 0.016}$ |
| | NUMAP-FT | $0.680_{\pm 0.334}$ | $0.348_{\pm 0.292}$ | $0.002_{\pm 0.001}$ | $0.635_{\pm 0.141}$ | $0.411_{\pm 0.153}$ | $0.272_{\pm 0.146}$ |

Table 10: An extension of Tab. 5 with the MNIST and FashionMNIST datasets. Global preservation results on MNIST are comparable between UMAP and P. UMAP, while superior to NUMAP-SN and NUMAP-FT. Regarding FashionMNIST, NUMAP, FashionMNIST, and NUMAP-SN are comparable, with NUMAP-FT having worse preservation.

Table 11: Technical details of the real-world datasets used for Sep-SpectralNet and NUMAP experiments.

| | Cifar10 | Appliances | KMNIST | Parkinsons | Wine | Banknote | MNIST | FashionMNIST |
|--------|---------|------------|--------|------------|------|----------|-------|--------------|
| #samples | 60,000 | 19735 | 70,000 | 5875 | 178 | 1372 | 60,000 | 60,000 |
| #features | 500 | 28 | 784 | 19 | 13 | 4 | 784 | 784 |

Table 12: Technical details in the Sep-SpectralNet experiments for all datasets.

| | Moon | Cifar10 | Appliances | KMNIST | Parkinsons |
|--------|------|---------|------------|--------|------------|
| Batch size | 2048 | 2048 | 2048 | 2048 | 512 |
| n_neighbors | 20 | 20 | 20 | 20 | 5 |
| Initial LR | $10^{-2}$ | $10^{-2}$ | $10^{-3}$ | $10^{-3}$ | $10^{-2}$ |
| Optimizer | ADAM | ADAM | ADAM | ADAM | ADAM |

As for the evaluation methods, we used a default choice of 5 neighbors to compute the kNN accuracy. The graph construction for GS is as detailed in App. C.2, using 50 neighbors to ensure connectivity.

For the computation of the p-values in Tab. 3 we used an independent t-test to compare the means of the 10 GS results, that were obtained from 10 runs using 10 different seeds.

**Time-series simulation.** We simulated two complex distributions in a 10-dimensional space. At each of the ten time steps, we sample a total of 5000 data points, 25% of which belong to the dynamic distribution (visualized by the pink dots in Fig. 7), while the other two distributions are kept the same. The dynamic distribution starts at the first (red) distribution, and linearly transitions into the other (blue). We used UMAP default parameters settings to visualize each time-step separately. As for NUMAP, we trained only on the first two time-steps, and obtained the others using a simple feed-forward operation.

We ran the experiments using GPU: NVIDIA A100 80GB PCIe; CPU: Intel(R) Xeon(R) Gold 6338 CPU @ 2.00GHz;

