# OpenReview forum: "Generalizable Spectral Embedding with an Application to UMAP"
_TMLR — Accepted by TMLR_

### Review · Reviewer_JvCb · 2025-05-12

**Summary Of Contributions:**

The paper proposes an extension of SpectralNet that is able to separate the eigenvectors while ensuring generalizability and scalability.  In addition, the paper also proposes a novel evaluation score, the *Grassmann score*, for better evaluating how well the global structure is preserved in graph-based dimensionality reduction methods.

**Audience:**

Yes

**Claims And Evidence:**

Yes

**Requested Changes:**

p.1: "These three properties are crucial for modern applications of SE in machine learning"
Why is this the case? Can you list easy-to-understand examples where all three properties are important? In particular, the last one (Eigenvector separation) seems to be quite technical, and it might not be clear for the general reader (incl. me) why this is an important property.
The examples in the following paragraph are not very detailed and not really convincing to me. Please elaborate more on this and assume less prior knowledge on the readers' side.

The same holds for "diffusion maps" in Section 4.4. I think a short statement on why they are interesting does not hurt.

p.2: "Our key contributions are: ..." I think the Grassman score is also a contribution, right? If so, why not list it there as well? This new evaluation score seems to be relevant for the community.

Some references have a DOI or a link, while others do not. I recommend being consistent, ideally having a DOI or link for every reference.

**Strengths And Weaknesses:**

The paper addresses a relevant problem and extends existing methods to close a research gap. The proposed method is supported by extensive empirical evaluations.

However, the readability and motivation of the paper could be improved to make it more accessible to a broader audience. I have concerns regarding the readability of the paper for readers that are not that deeply rooted in the paper's topic -- people like me.
I think it would be beneficial if the authors could better explain the meaning and the importance of eigenvector separations, which is their central contribution. The same holds for some other aspects (see "Requested Changes" for more detailed comments)

Please note that I do not have any background in this paper's topic. Maybe things are "obvious" to readers having a background in this topic.  I would be willing to step back from my requests if I get convinced that things are already pretty clear.

---

> ### Author Response · Authors · 2025-06-24
>
> We thank the Reviewer for their thoughtful feedback and for highlighting the importance of clarity and accessibility. We are especially grateful that you found our empirical evaluation extensive. Below, we address your specific points:
>
> **p.1:**
> We appreciate the concern regarding the clarity of this technical concept. In the revised version, we have expanded Section 1 to include a more accessible explanation of why eigenvector separation matters. In intuitive terms, well-separated eigenvectors ensure that the low-dimensional representation derived from spectral methods (e.g., spectral clustering, Diffusion Maps) is stable and meaningful.
>
> To make this more concrete, separated eigenvectors enable meaningful time-scale decompositions of the data in Diffusion Maps. To name a few, Diffusion Maps is widely used to analyze the transitions of molecules [1, 2], sample rare protein transitions [3], reveal the hidden structure of disordered materials [4], analyze the global arrangements of cortical features in several neuropsychiatric conditions [5,6,7], guide enhanced-sampling methods [8,9], and as dimensionality reduction in self-organizing networks [10]. We now clarify these points in the main text and include references for readers unfamiliar with the method.
>
> Furthermore, the empirical ablation in Appendix E demonstrates that eigenvector separation is a core enabler of our method (Tables 4 and 5). As a concrete example, we show below a comparison on CIFAR10 between NUMAP-SN (using SpectralNet) and NUMAP (using Sep-SpectralNet):
>
> |          | kNN ↑         | GS ↓              |
> | -------- | ------------- | ----------------- |
> | NUMAP-SN | 0.903 ± 0.002 | 0.281 ± 0.357     |
> | NUMAP    | 0.905 ± 0.004 | **0.051** ± 0.194 |
>
> These results highlight that the embedding quality, particularly in terms of global structure preservation, improves significantly with Sep-SpectralNet. We have added these insights to the revised version.
>
> **p.2:** Thank you for noticing this. We agree that the Grassmann Score is an important contribution and have added it to the list of contributions in the introduction.
>
> **Refrences.** Thank you for this. We have updated our bibliography to consistently include DOIs for all references where available.
>
> [1] Aldo Glielmo et al., Unsupervised learning methods for molecular simulation data. Chemical Reviews, 2021.
>
> [2] Eliodoro Chiavazzo et al., Intrinsic map dynamics exploration for uncharted effective free-energy landscapes. PNAS, 2017.
>
> [3] Danial Ghamari et al., Sampling a rare protein transition using quantum annealing. Journal of Chemical Theory and Computation, 2024.
>
> [4] Thomas J Hardin et al., Revealing the hidden structure of disordered materials by parameterizing their local structural manifold. Nature communications, 2024.
>
> [5] Bo-yong Park et al., Differences in subcortico-cortical interactions identified from connectome and microcircuit models in autism. Nature communications, 2021.
>
> [6] Bo-yong Park et al., Multiscale neural gradients reflect transdiagnostic effects of major psychiatric conditions on cortical morphology. Communications biology, 2022.
>
> [7] Debo Dong et al., Compression of cerebellar functional gradients in schizophrenia. Schizophrenia bulletin, 2020.
>
> [8] Wenwei Zheng et al., Delineation of folding pathways of a β-sheet miniprotein. The Journal of Physical Chemistry B,2011
>
> [9] Wenwei Zheng et al., Rapid exploration of configuration space with diffusion-map-directed molecular dynamics. The journal of physical chemistry B, 2013.
>
> [10] Paulo Valente Klaine et al., A survey of machine learning techniques applied to self-organizing cellular networks. IEEE Communications Survey & Tutorials, 2017.

---

> > ### Comment · Reviewer_JvCb · 2025-07-01
> >
> > Thanks for the clarification and the revision. This resolves all my concerns.

---

> > > ### Author Response · Authors · 2025-07-03
> > >
> > > Thank you. We appreciate you feedback.

---

### Review · Reviewer_ywS6 · 2025-05-16

**Summary Of Contributions:**

This paper presents a simple approach to recover approximations of the top k eigenvectors of a graph's laplacian matrix.  The approach performs an eigendecomposition on the output of a trained SpectralNet, evaluated on several minibatches of data, in order to recover the eigenvectors.  The main contribution of this paper is the application of this approach for improving parametric UMAP and constructing a new metric for evaluating umap embeddings.  This new UMAP algorithm preserved the global structure of data better than existing approaches, scales to large datasets, and is easy to implement.

**Audience:**

Yes

**Broader Impact Concerns:**

There are no broader impact concerns.

**Claims And Evidence:**

Yes

**Requested Changes:**

The technical details of the proposed approach (performing an eigendecomposition of the output of SpectralNet) should be added to the abstract or introduction clarify how exactly your approach relates to previous approaches.  I also think that the exposition should be written more from an application standpoint to highlight that this simple method can be used to substantially improve existing algorithms, as opposed to claiming a novel variant of SpectralNet.

Minor changes:
- Remove the word "recently" when referencing Shaham et al 2018 at top of page 2.
- Explain what A, m and X are in definition 1.
- Before equation 2 on page 7, write SE(X)_{1:l} to denote that Y uses the top l eigenvectors because SE(X) returns the top k.

**Strengths And Weaknesses:**

Strengths:
- The paper is well written and easy to understand.
- The problem is well motivated.
- The empirical results seem to indicate that the NUMAP algorithm does indeed preserve global structure better than existing approaches.

Weaknesses:
- The proposed technical contribution of finding the top eigenvectors seem obvious.  It does not seem like this paper adds much on a technical level to SpectralNet.
- The evaluation of SpectralNet's ability to recover the top eigenvectors seems uninformative because it is not designed to recover these eigenvectors.
- I'm not sure if the proposed method is required to construct NUMAP.  I would think that using SpectralNet to learn an l-dimensional representation directly and then using that to construct the NUMAP embedding should (in theory) yield the same results as first learning a k-dimensional representation and then taking its top l eigenvectors.

---

> ### Author Response · Authors · 2025-06-24
>
> We thank the Reviewer for their thoughtful and detailed comments, and for highlighting the strengths of our paper, including its clarity, motivation, and empirical validation of NUMAP. Below, we address each point.
>
> **Technical Contribution.** The key technical contribution lies in designing a pipeline that achieves eigenvector separation while maintaining the scalability and generalizability of SpectralNet. To our knowledge, no prior work has proposed this combination. This enables Spectral Embedding to satisfy all three desired properties simultaneously: generalizability, scalability, and eigenvector separation.
>
> Additionally, this separation allows a principled construction of the NUMAP embedding. We demonstrate how applying Sep-SpectralNet as a preprocessing step allows NUMAP to preserve global structure better than the existing approach. This combination is not merely a tweak, but a meaningful refinement motivated by spectral theory and confirmed through extensive experiments.
>
> **Eigenvectors Separation's Evaluation.** We agree that SpectralNet is not explicitly designed to recover the top eigenvectors. However, this is precisely the challenge we highlight. SpectralNet’s output does not correspond to separated eigenvectors. Our evaluation highlights this limitation and empirically validates that Sep-SpectralNet successfully achieves the intended separation.
>
> **Sep-SpectralNet is Required for NUMAP.**
> As noted above, SpectralNet learns an embedding without any guarantee that its axes correspond to specific eigenvectors, nor that they are ordered by importance. As such, truncating SpectralNet’s output from k dimensions to $\ell$ is not equivalent to selecting the top eigenvectors. In contrast, Sep-SpectralNet explicitly constrains separation (and thereby ordering), so truncation corresponds meaningfully to spectral decomposition.
>
> We demonstrate this in detail in Appendix E (Tables 4 and 5). As a concrete example, we show below a comparison on CIFAR10 between NUMAP-SN (using SpectralNet) and NUMAP (using Sep-SpectralNet):
>
> |          | kNN ↑         | GS ↓              |
> | -------- | ------------- | ----------------- |
> | NUMAP-SN | 0.903 ± 0.002 | 0.281 ± 0.357     |
> | NUMAP    | 0.905 ± 0.004 | **0.051** ± 0.194 |
>
> These results highlight that the embedding quality, particularly in terms of global structure preservation, improves significantly with Sep-SpectralNet.
>
> **Exposition.** In response to the Reviewer's comment, we revised the introduction to clarify the technical approach and emphasize its practical value. We now explicitly state that our method applies a post-processing eigendecomposition step to SpectralNet’s output, and we frame our contribution more from an application standpoint.
>
> **Minor Changes.** Thank you for pointing these out. We have implemented them in the revised version.

---

> > ### Author Response · Authors · 2025-07-30
> > **Following up comment on the rebuttal and review**
> >
> > We thank again the Reviewer for their feedback.
> >
> > We hope that our rebuttal has adequately addressed their concerns, and we kindly request for feedback on this, particularly in light of the positive responses to the rebuttal from Reviewers TLxe, JvCb.

---

### Review · Reviewer_TLxe · 2025-06-12

**Summary Of Contributions:**

In this work, the authors propose a dimensionality reduction method with three key properties: (1) out-of sample generalization, (2) scalability, and (3) edge vector separation (i.e., basis identification). The proposed method is an extension of the SpectralNet, which naturally achieves (1) and (2), by means of a post-processing step that achieves (3). They also propose an application to data visualization.

**Audience:**

Yes

**Claims And Evidence:**

No

**Requested Changes:**

- Critical: PCA discussion
- Critical: show why finding the exact basis is of critical importance
- Critical: Ablation study on skip connections in NUMAP
- Critical: the empirical section should cover scalability generalization thoroughly
- Critical: Discussion of the hyperparameter t in the main body of the paper and its effect on the metric.

**Strengths And Weaknesses:**

Strengths:
- The proposed Sep-SpectralNet is a conceptually straightforward extension of SpectralNet. I see this as a strength, in the sense that it can be more easily adopted than a more intríncate alternative.
- Although limited, the experimental results seem convincing.

Weaknesses:
- It seems that  the proposed SE approximation technique is very close to PCA (or an SVD, to be more specific) applied to the vectors f(x). From Equation (1), f(X)^T f(X) \approx L. Hence, the SVD of  f(X) approximates the eigendecomposition of L. I would appreciate a detailed discussion of this point.
- It is unclear why identifying the exact basis helps downstream applications. For clustering and visualization, rotating the input embeddings will not provide meaningful improvements from a practical point of view. Can the authors provide a setting where finding the exact basis is of critical importance?
- The comparison of Sep-SpectralNet with direct eigendecompositions of L in terms of runtime (Figure 2b) feels a bit unfair for a number of reasons. First, when comparing an approximate method to an exact one (up to numerical precision), one would normally analyze the relationship between error and runtime. Second, the authors should include Nystrom in that comparison, as it offers a significant acceleration over direct eigensolvers. From this perspective, and in combination with my previous point, it would be interesting to see a comparison between Sep-SpectralNet and Nystrom in terms of their errors.
- Why are skip connections an important part of the architecture? The authors should include an ablation study that shows that skip connections are useful.
- In a related but different line of work, Random Fourier Features RFFs rely on high-dimensional embeddings to alleviate the computational demands of downstream machine learning methods. These features, through Bochner’s theorem, are such that f(X)^T f(X) \approx K. Since Problem (1) can be posed as a maximization of A \^T K A, RFFs should work in theory. Of course, these features increase the dimensionality instead of reducing it, but their use has found a lot of success for many machine learning problems (for clustering, for example). Moreover, it seems like these features could be used as inputs to a NN minimizing the UMAP loss, just as NUMAP.  Of course, this approach would preclude using skip connections, but this may be cirtical or not (see my previous point).
- From the abstract, I was expecting to find large-scale experiments with millions of points actually shoing the scalability of the approach. There are many (labeled) datasets today at these scales for this evaluation. Since this is not a paper with strong theoretical results, the empirical section should be very strong and thorough.
- From the abstract, I was expecting a strong evidence of out-of-sample generalization by studying the relationship between the number of samples used for training and the generalization error. That is, an experiment that shows that the test error behaves nicely as function of the number of training samples.
- The authors train NUMAP in a stage-wise fashion. Since SVDs have been incorporated before in networks, they do not preclude end-to-end training. What would be the effect of this change?
- Finally, it seems to me that the hyperparameter t in GS has a fundamental effect on its values. For example, if multiple clusters are present in the data, a higher value of t would be needed to correctly recover these structures. I suggest including GS for multiple values of t in each experiment.

- Question: what is GrEASE in Figure 9 of the appendix?

---

> ### Author Response · Authors · 2025-06-24
> **Response to Reviewer RLxe (Part 1/2)**
>
> We thank the Reviewer for their thoughtful and detailed comments, which certainly improved the paper. We address your comments in detail below, and add the main insights to the revised version.
>
> **Connection to PCA.**
> We agree with the Reviewer that there is an underlying connection between our method and SVD. However, the key distinction lies in what the SVD is applied to. In our case, we apply SVD to the matrix $\tilde{\Lambda} = Q^T\Lambda Q$, where $\Lambda$ is the diagonal eigenvalues matrix of the Laplacian. This differs from PCA/SVD, which directly analyzes the covariance of raw inputs or features.
>
> Additionally, as noted on page Eq. 1, $f(X)^Tf(X) = I$ due to the orthogonality constraint imposed during SpectralNet training. This means that applying SVD to $f(X)$ is not meaningful in the PCA sense - it simply retrieves the identity. Our approach constructs a meaningful approximate eigenspace of the Laplacian using the learned embedding and then rotates it to approximate eigenvector alignment. This subtle but important distinction allows us to separate and rank eigenvectors, which PCA or naive SVD on $f(X)$ would not achieve.
>
> Of course, our method includes SVD in its core, but on a non-trivial choice of matrix. In particular, as written in page 6, retrieving the projection $Q$ by running SVD on $\tilde{\Lambda} = Q^T\Lambda Q$, which unlike $Q$ is computable. As stated in Eq. 1, $f(X)$ itself approximates the eigendecomposition of $L$ (minimizes the Rayleigh quotient), and $f(X)^Tf(X) = I$.
>
> **Motivation for Eigenvector Separation.**
> We agree that rotation-invariant downstream tasks like clustering or visualization can work with any orthogonal basis. However, there are several important cases where identifying specific eigenvectors is critical.
>
> One major use case is Diffusion Maps, where the eigenvectors and eigenvalues of the graph Laplacian define the diffusion geometry of the data. Computing the eigenvalues and matching them with their corresponding eigenvectors require eigenvectors separation. Diffusion maps has been vastly applied to analyze the transitions of molecules [1, 2], sample rare protein transitions [3], reveal the hidden structure of disordered materials [4], analyse the global arrangements of cortical features in several neuropsychiatric conditions [5,6,7], guide enhanced-sampling methods [8,9], and as dimensionality reduction in self-organizing networks [10].
>
> Additionally, the separation is the core idea that enables NUMAP. We demonstrate this in detail in Appendix E (Tables 4 and 5). As a concrete example, we show below a comparison on CIFAR10 between NUMAP-SN (using SpectralNet) and NUMAP (using Sep-SpectralNet):
>
> |          | kNN ↑         | GS ↓              |
> | -------- | ------------- | ----------------- |
> | NUMAP-SN | 0.903 ± 0.002 | 0.281 ± 0.357     |
> | NUMAP    | 0.905 ± 0.004 | **0.051** ± 0.194 |
>
>
> These results highlight that the embedding quality, particularly in terms of global structure preservation, improves significantly with Sep-SpectralNet. We have added these insights to the revised version.
>
> **Fig. 2b.** We appreciate the Reviewer’s observation. We first want to clarify that the scalability and generalizability of the method are inherited from SpectralNet, and the presented extension preserves them. Regarding the comparison, the goal of Fig. 2b is to illustrate that Sep-SpectralNet maintains scalability similar to existing (not generalizable) eigensolvers. Thereby, Nystrom-based methods are orthogonal to our goal in this figure (scalability) and not directly comparable in this setting.
>
> **Skip Connections Ablation Study.** We thank the Reviewer for this suggestion. Following the Reviewer's comment, we conducted an ablation study on the necessity of the skip connections on the CIFAR10 dataset. The results are shown in the following tables:
>
> |                             | kNN ↑         | GS ↓              |
> | --------------------------- | ------------- | ----------------- |
> | NUMAP (no skip connections) | 0.907 ± 0.003 | 0.191 ± 0.293     |
> | NUMAP                       | 0.905 ± 0.004 | **0.051** ± 0.194 |
>
> While local structure (kNN) is preserved without skip connections, the skip connections significantly improve global structure preservation, indicated by the lower GS. We have incorporated these new results in the revised version.
>
> **Random Fourier Features.** We thank the Reviewer for this suggestion. Exploring alternative inputs to the UMAP loss network, such as RFFs, is an interesting direction for future work and is now mentioned in the Discussion of the revised version.

---

> > ### Author Response · Authors · 2025-06-24
> > **Response to Reviewer TLxe (Part 2/2)**
> >
> > **Scalability and Generalizability.** Sep-SpectralNet extends SpectralNet [11], in the sense that is adds eigenvector separation while preserving its other properties. In particular, Sep-SpectralNet inherits the scalability and generalizability of SpectralNet. The added eigenvector separation step involves only a $k\times k$ matrix diagonalization, which has negligible cost relative to the dataset size. Thus, our method retains SpectralNet’s core advantages while enhancing its spectral properties.
> >
> > **NUMAP Training.** We indeed trained NUMAP in two stages: first, Sep-SpectralNet is trained to learn a spectral embedding; then, a separate neural network is trained using the UMAP loss. We considered two alternatives for end-to-end training. The first is Parametric UMAP, which skips the spectral step and thus fails to preserve global structure - this is reflected in its lower performance in GS in Table 3. The second is fine-tuning Sep-SpectralNet with the UMAP loss (NUMAP-FT). However, as shown in Appendix E, this approach also performs worse than our two-stage method on both local and global structure metrics. For instance, results on CIFAR10 are provided in the table below.
> >
> > |          | kNN ↑             | GS ↓              |
> > | -------- | ----------------- | ----------------- |
> > | NUMAP-FT | 0.577 ± 0.081     | 0.680 ± 0.334     |
> > | NUMAP    | **0.905** ± 0.004 | **0.051** ± 0.194 |
> >
> >
> > **GS Hyperparameter.** We thank the Reviewer for pointing this out. As discussed in Sec. 4.5, the number of eigenvectors t affects the emphasis between global and local structures. For consistency and interpretability, we report GS using only the Fiedler vector (t=2) in all experiments, while using a sufficiently large number of neighbors to ensure the connectivity of the graph. This choice avoids tuning a hyperparameter and aligns with standard graph learning, where the Fiedler vector often captures meaningful global structure. We have clarified this in the revised version.
> >
> > **Fig. 9** Thank you for noticing it. We have corrected this in the revised version.
> >
> > [1] Aldo Glielmo et al., Unsupervised learning methods for molecular simulation data. Chemical Reviews, 2021.
> >
> > [2] Eliodoro Chiavazzo et al., Intrinsic map dynamics exploration for uncharted effective free-energy landscapes. PNAS, 2017.
> >
> > [3] Danial Ghamari et al., Sampling a rare protein transition using quantum annealing. Journal of Chemical Theory and Computation, 2024.
> >
> > [4] Thomas J Hardin et al., Revealing the hidden structure of disordered materials by parameterizing their local structural manifold. Nature communications, 2024.
> >
> > [5] Bo-yong Park et al., Differences in subcortico-cortical interactions identified from connectome and microcircuit models in autism. Nature communications, 2021.
> >
> > [6] Bo-yong Park et al., Multiscale neural gradients reflect transdiagnostic effects of major psychiatric conditions on cortical morphology. Communications biology, 2022.
> >
> > [7] Debo Dong et al., Compression of cerebellar functional gradients in schizophrenia. Schizophrenia bulletin, 2020.
> >
> > [8] Wenwei Zheng et al., Delineation of folding pathways of a β-sheet miniprotein. The Journal of Physical Chemistry B,2011
> >
> > [9] Wenwei Zheng et al., Rapid exploration of configuration space with diffusion-map-directed molecular dynamics. The journal of physical chemistry B, 2013.
> >
> > [10] Paulo Valente Klaine et al., A survey of machine learning techniques applied to self-organizing cellular networks. IEEE Communications Survey & Tutorials, 2017.
> >
> > [11] Uri Shaham et al., Spectralnet: Spectral clustering using deep neural networks. ICLR, 2018.

---

> > > ### Comment · Reviewer_TLxe · 2025-07-12
> > >
> > > The above answers address most of my concerns. Thanks for the detailed response.

---

### Author Response · Authors · 2025-06-24

Dear Reviewers and AE,

We thank the Reviewers and the Area Chair for their time, thoughtful feedback, and constructive suggestions. We are encouraged by the overall positive reception and appreciation for the clarity, motivation, and empirical strength of our work. In the attached revised version, we have addressed all major comments, clarified technical points, added new ablation studies, and expanded relevant discussions accordingly. Changes are colored in blue. We hope these improvements further strengthen the manuscript.

Thank you

---

### Decision · Action_Editor_YCdH · 2025-07-28

**Recommendation:** Accept with minor revision

**Additional Comments:**

Reviewers had concerns about how the paper advertises 'scalability', e.g., in the abstract
> We empirically demonstrate Sep-SpectralNet's ability to consistently approximate and generalize SE, while maintaining scalability.
However, the conducted experiments are not conducted on what would currently be considered large datasets. While the method appears to scale similarly to other spectral methods, this does not imply that it is scalable to extremely large datasets. Please revise the paper to either (1) make it clear that the method is as scalable as the methods it builds upon, or (2) demonstrate empirically that the method is truly scalable.

**Audience:**

Yes

**Audience Explanation:**

The paper focuses on dimensionality reduction in the style of spectral embeddings. Such are commonly used for visualization. There is a small but active community working on this style of method.

**Claims And Evidence:**

Yes

**Claims Explanation:**

In general, the paper lives up to its claims. It points to a series of issues with spectral embeddings, proposes remedies, and empirically demonstrates that they improve the situation.

One exception is that the paper is imprecise about the term 'scalability', which the authors have been requested to reword in a minor revision.

---

> ### Author Response · Authors · 2025-07-31
> **Authors' Response to Minor Revisions**
>
> Thank you for the constructive feedback and for the positive decision.
>
> We revised the paper in line with the AE's request. Along other minor refienments, we clarified the use of the term scalability in the abstract and updated the caption of Fig. 2b to better reflect the method's scalability relative to SpectralNet (the key prior spectral method).
>
> The updated camera-ready version is attached.
>
> We appreciate the thoughtful reviews and the helpful guidance throughout the process.